# Highly Organized Porous Gelatin-Based Scaffold by Microfluidic 3D-Foaming Technology and Dynamic Culture for Cartilage Tissue Engineering

**DOI:** 10.3390/ijms23158449

**Published:** 2022-07-30

**Authors:** Hsia-Wei Liu, Wen-Ta Su, Ching-Yi Liu, Ching-Cheng Huang

**Affiliations:** 1Department of Life Science, Fu Jen Catholic University, New Taipei City 242062, Taiwan; 079336@mail.fju.edu.tw (H.-W.L.); 079336@gmail.com (C.-Y.L.); 2Graduate Institute of Applied Science and Engineering, Fu Jen Catholic University, New Taipei City 242062, Taiwan; 3Graduate Institute of Biochemical and Biomedical Engineering, National Taipei University of Technology, Taipei 106344, Taiwan; f10549@mail.ntut.edu.tw; 4Department of Biomedical Engineering, Ming-Chuan University, Taoyuan 333321, Taiwan; 5PARSD Biomedical Material Research Center, Taichung 407428, Taiwan

**Keywords:** microfluidics, gelatin, cartilage tissue engineering, dynamic culture

## Abstract

A gelatin-based hydrogel scaffold with highly uniform pore size and biocompatibility was fabricated for cartilage tissue engineering using microfluidic 3D-foaming technology. Mainly, bubbles with different diameters, such as 100 μm and 160 μm, were produced by introducing an optimized nitrogen gas and gelatin solution at an optimized flow rate, and N_2_/gelatin bubbles were formed. Furthermore, a cross-linking agent (1-ethyl-3-(3-dimethyl aminopropyl)-carbodiimide, EDC) was employed for the cross-linking reaction of the gelatin-based hydrogel scaffold with uniform bubbles, and then the interface between the close cells were broken by degassing. The pore uniformity of the gelatin-based hydrogel scaffolds was confirmed by use of a bright field microscope, conjugate focus microscope and scanning electron microscope. The in vitro degradation rate, mechanical properties, and swelling rate of gelatin-based hydrogel scaffolds with highly uniform pore size were studied. Rabbit knee cartilage was cultured, and its extracellular matrix content was analyzed. Histological analysis and immunofluorescence staining were employed to confirm the activity of the rabbit knee chondrocytes. The chondrocytes were seeded into the resulting 3D porous gelatin-based hydrogel scaffolds. The growth conditions of the chondrocyte culture on the resulting 3D porous gelatin-based hydrogel scaffolds were evaluated by MTT analysis, live/dead cell activity analysis, and extracellular matrix content analysis. Additionally, a dynamic culture of cartilage tissue was performed, and the expression of cartilage-specific proteins within the culture time was studied by immunofluorescence staining analysis. The gelatin-based hydrogel scaffold encouraged chondrocyte proliferation, promoting the expression of collagen type II, aggrecan, and sox9 while retaining the structural stability and durability of the cartilage after dynamic compression and promoting cartilage repair.

## 1. Introduction

Osteoarthritis (OA) is one of the most common orthopedic conditions, generally affecting patients over 50 years old, with symptoms such as joint pain and decreased function [1]. There are two ways to treat osteoarthritis. One is through conservative drug treatment, which is mainly used to suppress pain and inflammation; the other is through the use of surgery. Chondrocytes are terminally differentiated cells with extremely weak proliferation capability and a lack of regenerative capability, thus, they are unable to self-repair after cartilage defect or injury. Therefore, cartilage defect or injury has been a major problem in surgical clinical treatment, and cartilage tissue engineering has been receiving more attention for clinical needs. In previous works, some materials have been proposed, modified, and employed for biomedical and regenerative applications, such as scaffolds for skin and bone tissue reconstruction [2,3,4,5,6,7]. Regenerative biomaterials provide the necessary substances to support the growth of cells and tissues, maintain the shape and mechanical properties of regenerated tissues, and promote integration with surrounding tissues. The surfaces and interfaces of biomaterials interact directly with cells and tissues, and thus, significantly influence many cellular behaviors such as adhesion, spreading, proliferation, migration, and differentiation, as well as the outcomes of tissue repair and the regeneration of cartilage. Cells were seeded in a three-dimensional scaffold constructed from biodegradable regenerative materials, which was expected to provide the mechanical and chemical signals required by cells to regulate cell adhesion, proliferation, and differentiation, and finally, to assemble themselves into three-dimensional tissues [8]. A porous scaffold with a three-dimensional microstructure was considered for applications of cartilage tissue engineering. However, a porous scaffold with uniform three-dimensional holes and high interconnectivity was difficult to prepare. For these reasons, several techniques, such as bioprinting procedures [9] and microfluidic procedures, were studied [10,11,12]. Particularly, some conventional microfluidic systems were designed and employed to fabricate scaffolds efficiently with a desirable homogenous porosity, uniform spatial structure, interconnectivity, and potential mechanical properties [10,11,12]. For application purposes, these conventional microfluidic techniques were applied to fabricate new materials that encapsulate target cells in small spherical porous foam, which would be a powerful method for tissue engineering [10,11,12].

Gelatin is a protein of animal origin, obtained by acid or base hydrolysis of collagen. Gelatin has excellent biocompatibility, biodegradability, non-immunogenicity, and a great capacity for modification at the amino acid level [13,14,15]. Due to gelatin’s superior properties, it can be considered and employed as a biomaterial for both hard and soft-tissue engineering and medical applications [13,14,15]. In general, a gelatin hydrogel can be formed by physical crosslinking in water above a certain concentration and below 30~35 °C [16,17]. Gelatin molecules aggregate and undergo a conformational change from a random coil to a triple helix [13,14,15]. Thus, gelatin hydrogels have low shape stability, poor mechanical strength, and low elasticity [18], which significantly limits their biomedical applications at physiological temperatures above 37 °C. To enhance its stability and mechanical properties, gelatin can be covalently crosslinked through the use of chemical crosslinking agents such as formaldehyde, glutaraldehyde, paraformaldehyde, 1-ethyl-3-(3-dimethylaminopropyl) carbodiimide hydrochloride (EDC), and genipin [19,20,21]. Additionally, the use of enzymes such as tyrosinase and transglutaminase can initiate an enzymatic crosslinking procedure, causing gelatin crosslinking, as studied by [22,23]. Other physical crosslinking methods, such as plasma treatment, often result in weak gelatin macromolecule crosslinking because crosslinking occurs only at the surface of the material [24]. Photocrosslinked gelatin-based hydrogels usually present short gelation time and are chemically stable and mechanically strong, but the photo-initiators required for the photopolymerization reaction might lead to cell death [25]. Furthermore, gelatin can also be cross-linked using microbial transglutaminase (MTG), an FDA-approved enzyme that covalently bonds the glutamine and lysine groups between gelatin polymers [26].

In this study, a microfluidic system was employed to prepare three-dimensional porous EDC-crosslinked gelatin-based hydrogel scaffolds with highly organized uniform porosity and pore size for use as tissue engineering scaffolds. The pore size and pore uniformity of the hydrogel scaffolds would be confirmed by a bright field microscope, conjugate focus microscope, and scanning electron microscope, respectively. The rabbit knee cartilage chondrocytes were cultured. Its extracellular matrix content, histological analysis, and immunofluorescence staining were analyzed. The rabbit knee chondrocytes were seeded on three-dimensional porous gelatin-based hydrogel scaffolds, and chondrocyte growth conditions were evaluated by the MTT method, by live/dead cell activity analysis, and by analyzing the extracellular matrix content. Furthermore, a dynamic culture of chondrocytes was applied to enhance cell growth. Additionally, the expression of cartilage-specific proteins was studied by immunofluorescence staining analysis in relation to the dynamic culture time to evaluate the highly organized cross-linked porous gelatin-based hydrogel scaffold with uniform open cells by microfluidic 3D-foaming technology and dynamic culture for cartilage tissue engineering.

## 2. Results and Discussion

### 2.1. Highly Organized EDC-Crosslinked Porous Gelatin-Based Hydrogel Scaffold with Uniform Open Cells

In general, a porous scaffold can be prepared by using physical treatments (such as crushing, grinding, compacting, crystallization, or freeze-drying, etc.). Various pore sizes and interconnected shapes would affect the scaffold’s physicochemical properties [27]. Previous studies have shown that if the pore size variation of three-dimensional tissue engineering scaffolds is larger, it will hinder cell seeding and proliferation [27].

In this study, the gelatin solution was introduced into the parallel microfluidic channels, and N_2_/gelatin bubbles of uniform size were produced after passing through the microfluidic channels (Figure 1A), and then the N_2_/gelatin bubbles were collected (Figure 1B), cooled (Figure 1C) and solidified by chemical cross-linking reaction (Figure 1D,E). After degassing to remove nitrogen, a hydrogel scaffold with uniform and interconnected pores (open-cells structure) could be formed (Figure 1E). The open-cells structure of the resulting hydrogel scaffold was further identified by a bright field microscope, a confocal microscope, and scanning electron microscopy (SEM), as shown in Figure 2. The interconnected porous structure of the hydrogel scaffolds with a pore size of 160 μm and 100 μm were observed. Highly organized regular arrangement of open cells were showed in both hydrogel scaffolds.

### 2.2. Swelling Property and Degradation Behavior of EDC-Crosslinked Gelatin-Based Hydrogel Scaffolds

Hydrogel is a network three-dimensional structure, which is composed of many hydrophilic polymers. Gelatin segments had some hydrophilic functional groups such as -OH, -COOH, and -CONH_2_, which could bind with water to form a water-containing skeleton microstructure by physical crosslinking behavior (Figure 3A). Maintaining the microstructure, these functional hydrophilic functional groups in the microstructure would provide high opportunities to capture large numbers of water molecules, enabling the hydrogel structure to maintain a relatively high proportion of water contents. A chemical crosslinking reaction could further enhance the strength of the hydrogel’s structure and provide the swelling property for gelatin-based hydrogel scaffolds (Figure 3B).

In general, degradation of a hydrogel scaffold might occur via surface erosion or bulk erosion, which would be dependent on the amount of incorporated degradable units, cross-link density, and access of water to the interior of the hydrogel [28,29,30]. Surface erosion would occur when water did not penetrate the interior of the hydrogel scaffold due to high cross-link density or limited access to cleavage points, forcing the surface or exterior bonds to cleave first. Bulk erosion would occur essentially homogeneously throughout the hydrogel, with interior and exterior bonds being cleaved simultaneously. Bulk erosion was common in hydrogels because of their high internal water content and fast diffusion [28,29,30]. In this study, gelatin-based hydrogel scaffolds with uniform open cells were designed and prepared successfully. The uniform open cells would enhance the degradation of bulk erosion. Faster clearance of the hydrogel would be beneficial in vivo, as quick degradation contributes to a reduced immune response. When designing a hydrogel for a specific biomedical application such as a scaffold for tissue engineering, the degradation rate is imperative for the function of the hydrogel and clinic needs. A relative low degradation rate might provide target cells the time to develop extracellular matrix and to reintegrate damaged tissue. It is important to balance the benefit of a hydrogel as a scaffold for developing cells and tissues with a reduced immune response. To study the decomposition behavior of a scaffold, the erosion behavior and degradation should be considered. When surface erosion of a scaffold occurs, the scaffold’s decomposition would begin at the surface, and the invasion speed gradually increases as the water molecules invade the interior of the scaffold. At the same time, the volume of the scaffold decreases with time from the erosion surface. When bulk erosion occurs, it means that the speed of the water molecules entering the interior of the carrier is faster than that at the beginning of the erosion, resulting in the hydrolysis of the macromolecules inside the scaffold. In this study, the volume of the hydrogel scaffold would decrease with time during the six weeks of decomposition. The gelatin-based hydrogel scaffold with uniform open cells exhibited high water contents, strong hydrophilicity, and a porous structure, which could allow water to enter the inner part of the scaffolds easily, thereby enhancing the erosion effect from the inside and outside of scaffolds.

Furthermore, degradation behavior of hydrogel scaffolds refers to the breaking of the cross-linked network structure inside the hydrogel scaffolds over time, which changes the physical, chemical, and mechanical properties of the hydrogel scaffolds. There are many factors that affect the degradation behavior, including the swelling ratio of the hydrogel, the water content at swelling equilibrium, and the mechanical properties of the network structure. The gelatin hydrogel was chemically cross-linked and then degraded at 37 °C. The hydrogel was collected, degraded, and lyophilized. After weighing, the weight loss rate was calculated, and the weight loss rate was plotted against time. Overall, the gelatin hydrogel decomposition rate reached a weight loss of 7% after 6 weeks. Comparing the decomposition rate of scaffolds, no significant difference between scaffolds with different pore sizes was observed (Figure 4A).

In general, when the crosslinked hydrogel was immersed in the aqueous solution, the water molecules would contact the surface of the hydrogel and penetrate into the polymeric network structure of the hydrogel, causing a gap to appear between the glassy phase parts of the hydrogel that had not contacted the aqueous solution and the rubbery phase part that had contacted the water molecules. A moving boundary would be formed, and the meshes of the rubbery phase network would begin to expand, allowing more water molecules to enter the hydrogel, which is called swelling. When the hydrogel reaches swelling equilibrium, the swelling degree no longer changes. When crosslinked hydrogels swell, the glassy phase turns into the rubbery phase [31]. The degree of crosslinking would affect the area permitted for diffusion across the crosslinked hydrogel network, as well as the capacity for the hydrogels to take up water [32]. In this study, highly organized pores with different diameters were introduced to the crosslinked hydrogel scaffolds. The highly organized pores would provide channels to promote the penetration of water molecules into the polymeric network structure of the hydrogel scaffold, thereby allowing for effective swelling. The swelling structure of porous gelatin-based hydrogel scaffolds with highly organized pores could provide a large area of hydrophilic surfaces for the promotion of cell growth. Comparing the swelling ratio of scaffolds with different pore sizes, the gelatin-based hydrogel scaffolds with uniform pores of 160 μm diameter showed a relatively high swelling ratio. The swelling ratios of gelatin-based hydrogel (7%) scaffolds with different pore sizes were observed in the range of 38–43%, as shown in Figure 4B. Similar swelling ratios were observed by Xing et al. and Wisotzki et al. for EDC-crosslinked gelatin hydrogels in a range of 20–60% and low crosslinked gelatin (<7%) in a range of 33–43%, respectively [33,34]. The tensile stress values of EDC-crosslinked gelatin hydrogels with different pore sizes were determined to be ca.10.8 and 8.4 for 100 μm and 160 μm, respectively, as shown in Figure 4C, which indicated the structural strength of the resulting EDC-crosslinked gelatin-based hydrogel scaffolds with different pore sizes (160 and 100 μm). Furthermore, the elongations at break of EDC-crosslinked gelatin-based hydrogel scaffolds with different pore sizes (160 and 100 μm) were observed in a range of 19–22%, indicating that the EDC-crosslinked gelatin-based hydrogel scaffolds with different pore sizes (160 and 100 μm) all exhibited flexibility (Figure 4C).

### 2.3. Immunofluorescence Staining of Rabbit Knee Cartilage Cells

In this study, an evaluation of a rabbit knee chondrocyte culture was carried out in vitro. An isolation and culture of cartilage cells from the knee joint of rabbits was conducted. After the cartilage fragments were treated with enzymes and aseptic, the chondrocytes began to crawl out of the cartilage fragments after 7 days (Figure 5A). The chondrocytes were in the shape of stars, triangles, and polygons. When the cells were at primary passage (P0), they began to divide and proliferate. After 7 days, the cells grew to almost fill the plate and were packed tightly (Figure 5B). Cells isolated from rabbit knee cartilage were labeled with type II collagen, aggrecan, MMP13, and Sox9 by fluorescent staining. The type II collagen, aggrecan, MMP13, and Sox9 were cartilage-specific proteins that could be stained for the identification of chondrocytes, as shown in Figure 5C–F.

Articular cartilage, including chondrocytes and extracellular matrix (ECM), are a special connective tissue able to endure high weight loads for a long time. The extracellular matrix of cartilage contains various mixtures such as proteoglycans (proteoglycans), hyaluronic acid (HA), type II collagen, glycoproteins, and elastic fibers. Collagen type II is the main protein in knee cartilage cells. As chondrocyte monolayers dedifferentiate over time, type I collagen and III collagen increase, while type II collagen decreases. Type II collagen could be used to identify the dedifferentiation status of cartilage tissues [35,36,37]. Matrix metalloproteinase 13 (MMP13, collagenase III) is a matrix metalloproteinase that plays an important role in endochondral ossification and bone remodeling. MMP13 could contribute to the decomposition and reconstruction of cartilage in degenerative arthritis. The amount of MMP13 in early degenerative arthritis and normal knee cartilage was significantly higher than that in advanced degenerative arthritis and chondrocytes [38]. As shown in Figure 5C,D, Type II collagen and MMP13 were observed by fluorescent staining, indicating the activity of isolated chondrocytes. Additionally, another important molecule in proteoglycans that enables the normal operation of articular cartilage is aggrecan, which can be polymerized by a large amount of chondroitin sulfate proteoglycans as a cartilage-specific protein indicator [39,40]. The observation of aggrecan fluorescent staining in cells isolated from rabbit knee cartilage indicates isolated chondrocyte activity (Figure 5E). Furthermore, Sox-9 played a crucial role in the specific activation of collagen type II, and was a key transcriptional regulator of chondrocyte differentiation, demonstrating it to be an important protein for chondrocyte expression [38,41,42]. Sox9 was observed by fluorescent staining, indicating chondrocyte expression, as shown in Figure 5F.

### 2.4. Cell viability and Cell Growth on EDC-Crosslinked Gelatin-Based Hydrogel Scaffolds

Chondrocytes were seeded into gelatin-based hydrogel scaffolds at a concentration of 1 × 10^6^ cell/scaffold. The cell viability and cell number could be observed by live/dead assay. Cell viability and cell growth could be observed by live/dead fluorescence staining. As shown in Figure 6, living cells were stained with green fluorescence and dead cells were stained with red fluorescence. The fluorescent images appeared yellow in the overlay of live and dead cells [43]. The fluorescent images showed live (green) chondrocytes cultured on gelatin-based hydrogel scaffolds over a static culturing period of 7 days. The total number of cells containing dead and live cells increased with the number of days. After static culturing, the chondrocytes were covered on the majority of the EDC-crosslinked gelatin-based hydrogel scaffolds’ surface over a period of 21 days, as shown in Figure 6C, which exhibited overcrowded chondrocytes on the scaffold surface and low interconnectivity. Furthermore, the cell viability and cell number could be detected by MTT assay. The MTT assay showed a clear decrease of the number of cells, indicating a decrease trend of cell activity. The active cell numbers decreased from 2.3 × 10^5^ cells (after 1 day) to 4.6 × 10^4^ cells (after 21 days). The chondrocyte-covered EDC-crosslinked gelatin-based hydrogel scaffolds with relatively large pore sizes of 160 μm might provide relative high interconnectivity for culture to promote cell migration.

Since chondrocytes normally secrete extracellular matrix, total collagen/DNA and the amount of GAG/DNA were the basis for judging the degree of cartilage of cells. As shown in Figure 7, the amount of total collagen/DNA of static culturing chondrocytes on gelatin-based hydrogel scaffolds increased with the increase of culture time, from about 5.0 (μg/μg) for 1 day to ca. 17.0 (μg/μg) for 21 days.

From the results in Figure 8, it can be seen that the amount of GAG/DNA secretion increases with the increase of culture time, and can rise from about 0. 2 on the first day to about 1.1(μg/μg) on the 21st day for the gelatin-based hydrogel scaffold with a pore size of 100 μm.

From the results of GAG/DNA secretion for 21 days, a GAG/DNA secretion of about 2.0 (μg/μg) was observed for the gelatin-based hydrogel scaffold with a 160-μm pore size and about 1.1 (μg/μg) for the gelatin-based hydrogel scaffold with a 100-μm pore size. The amount of GAG/DNA secretion was significantly different between the hydrogel scaffolds with a pore size of 160 μm and a pore size of 100 μm.

After the static culturing, the results of MTT and DNA analysis showed that the number of cells decreased with the culture time. When the chondrocytes were seeded into the homogeneous hydrogel scaffold, the migration of the cells was slow, the chondrocytes stayed on the surface of the scaffold, and did not have enough space to grow any more cells. When the growth space is insufficient, the cells will begin to age and die gradually, and the nutrients and metabolic wastes of the culture medium in static culture cannot be immediately replaced, resulting in the death of the chondrocytes or weak chondrocyte activity.

### 2.5. Scanning Electron Microscope Analysis of Static Culturing Chondrocytes on EDC-Crosslinked Gelatin-Based Hydrogel Scaffolds

After static culturing chondrocytes on gelatin-based hydrogel scaffolds for 7 days and 14 days, the morphology and distribution of cells were observed by scanning electron microscope, as shown in Figure 9. Chondrocytes were evenly distributed on the surface, and the pores of the gelatin-based hydrogel scaffold were covered (Figure 9A,B). After 14 days, the chondrocytes could not migrate to the center of the scaffold, but rather, they stayed on the surface of the scaffold (Figure 9C,D).

### 2.6. Characteristics of Dynamic Culturing Chondrocytes on EDC-Crosslinked Gel-Atin-Based Hydrogel Scaffolds

To promote cell migration and the proliferation of chondrocytes, the exchange rate of culture medium and gas in the gelatin-based hydrogel scaffold must be promoted. The gelatin-based hydrogel scaffold with a pore size of 160 μm in diameter was dynamically cultured for 7 days, 14 days, 21 days, and 28 days, respectively, and the growth distribution and morphology of the chondrocytes were observed by scanning electron microscope, as shown in Figure 10. The EDC-crosslinked gelatin-based hydrogel scaffold with uniform pores of 160 μm in diameter was dynamically cultured with chondrocytes for 7 days (Figure 10A), and the cells were mainly distributed on the surface. Viewed from the cross section of the scaffold, the cells resided in the first to second layers (Figure 10B). Furthermore, the dynamic culture was performed for 14 days, and the distribution of cells on the scaffold surface gradually decreased (Figure 10B). Viewed from the cross section of the scaffold with uniform pores of 100 μm in diameter, there were not any cells inside the scaffold after static culture for 14 days, as shown in Figure 9C. Even on the scaffold with uniform pores of 100 μm in diameter, it is difficult to find cells inside the scaffold after static culture for 14 days, as shown in Figure 9D. The interconnectivity of chondrocytes static cultured on a scaffold affects the migration and growth of cells. Even increasing the time of static culture, chondrocytes might find it difficult to migrate to the center of the scaffold because the cultures become overcrowded on the surface of the scaffold, as proposed in Figure 11C. After 21 days of dynamic culture, there were significantly fewer cells on the scaffold surface than at 7 days (Figure 10A,E). Viewed from the cross section of the scaffold, cells had migrated to the first and second layers and had begun to divide and proliferate (Figure 10F). Because poor migration of cells was observed after static culture for 14 days, static culture was displaced with dynamic culture to promote the migration of cells. Poor cell migration after static culture might be exhibited, and good migration of cells after dynamic culture could be observed for 28 days. Viewed from the cross section of the scaffold, chondrocytes had been migrated, uniformly distributed, and proliferated in the EDC-crosslinked gelatin-based hydrogel scaffold after 28 days of dynamic culture (Figure 11A). Comparison between static and dynamic culture were proposed as shown in Figure 11B,C, respectively.

Furthermore, a DNA content assay was studied. As the number of culturing days increased, the DNA content increased from ca.1.4 μg/mL for 7 days to ca.2.4 μg/mL for 28 days, which exhibited that the cells would increase with culturing days, as shown in Figure 12A. As shown in Figure 12B, the total collagen/DNA of dynamic cultured chondrocytes on gelatin-based hydrogel scaffolds increased with the increase of culture time, from about 15 μg/μg for 1 day to ca.65 μg/μg for 21 days. Remarkable changes were observed. The total collagen/DNA of dynamic cultured chondrocytes on gelatin-based hydrogel scaffolds was reduced for 28 days, which might be due to a lack of sufficient oxygen and nutrients needed to support cell viability on an overcrowded scaffold. From the results in Figure 12C, it can be seen that the amount of GAG/DNA secretion increases with the increase of culture time, and can rise from about 3.8 on the first day to about 4.2 (μg/μg) on the 21st day. A decreased GAG/DNA value was observed at 28 days, which might be similarly due to a lack of sufficient oxygen and nutrients needed to support cell viability on an overcrowded scaffold.

Chondrocytes maintain articular cartilage through coordinated production and degradation of the extracellular matrix. Rabbit chondrocytes were isolated from articular cartilage and dynamic cultured EDC-crosslinked gelatin-based hydrogel scaffold. Figure 13 showed the immunofluorescence staining of expression from chondrocytes grown on EDC-crosslinked gelatin-based hydrogel scaffold with pore sizes of 160 μm diameter after 21 and 28 days of dynamic culture for collagen type II, MMP13, sox9, and aggrecan. From Figure 13A,B, it can be observed that the content of type II collagen gradually increased after 28 days. Type II collagen is one of the main proteins comprising knee cartilage. As chondrocyte monolayer cultures de-differentiated over time, collagen types I and III increased, while collagen type II decreased. After dynamic culture, the number of cells and type II collagen increased, indicating that differentiation of chondrocytes was carried out on collagen type II increases, as shown on the EDC-crosslinked gelatin-based hydrogel scaffold (160 μm). 

Figure 13C,D showed the immunofluorescence staining results of matrix metalloproteinase 13 (MMP13) after dynamic culture of gelatin-based hydrogel scaffold (160 μm) and chondrocytes for 21 days and 28 days. According to the results, MMP13 expression increases with the culture time. Figure 13E,F showed the sox9 expression results of immunofluorescence staining form chondrocytes grown on a gelatin-based hydrogel scaffold (160 μm) of dynamic culture after 21 days and 28 days. Sox9 maintained its expression after 21 days and 28 days of culture. Figure 13G,H showed the aggrecan immunofluorescence staining analysis of EDC-crosslinked gelatin-based hydrogel scaffold (160 μm) and chondrocytes after 21 days and 28 days of dynamic culture. Aggrecan maintained the expression after 21 days and 28 days of culture. According to the results of fluorescence staining, chondrocytes were seeded on an EDC-crosslinked gelatin-based hydrogel scaffold (160 μm) via dynamic culture and maintained the appearance of the cartilage. In the tidal bioreactor for dynamic culture, chondrocytes had good growth without dedifferentiation.

The results of the gene expression analysis showed that dynamic culture and uniform porous scaffolds could increase the expression of extracellular matrix genes in chondrocytes. Collagen type II, MMP13, sox9, and aggrecan are all cartilage-specific expression genes, and the expression levels of these genes are different. The EDC-crosslinked gelatin-based hydrogel scaffold encouraged chondrocytic proliferation, promoting the expression of collagen type II, aggrecan, and sox9, while retaining the mechanical strength and durability of the cartilage after dynamic compression and promoting cartilage repair.

Histological analyses such as hematoxylin and eosin staining are common staining methods. The staining results showed that the nucleus was blue, and the cytoplasm, intercellular substance, and various fibrous substances were light red to red. As shown in Figure 14, chondrocytes located at the scaffolds were observed after seven days of dynamic cell culture. However, the chondrocytes were only distributed on the near surface part of the gelatin-based hydrogel scaffold. After seven days of dynamic cell culture, chondrocytes grew into the scaffold, which revealed that more cells existed in the 3D construct. Furthermore, alcian blue easily combined with acidic mucus and polysaccharides, and alcian blue staining could show GAG. Because the dye could react with sulfur groups, it is blue in positive reaction, and the nucleus is red. After seven days of cell culture, most of the chondrocytes were distributed on the surface of the gelatin scaffold, and the produced GAG were also distributed on the surface of the scaffold, as shown in Figure 15. Chondrocytes would grow inside the scaffold for 7 days. After 28 days of dynamic cell culture, the cells grew uniformly in the hydrogel scaffold, but GAG in the cartilage tissue decreased. This result showed the same trend as GAG/DNA secretion, which showed that the chondrocyte extracellular matrix would decrease after 28 days. The dynamic culture could improve cell proliferation, which was due to the good interconnectivity in the dynamic culture system.

## 3. Materials and Methods

### 3.1. Materials

All the chemicals, such as Type A gelatin (Sigma-Aldrich, St. Louis, MO, USA), Pluronic F127 (Invitrogen, Carlsbad, CA, USA), perfluorohexane (Merck, Darmstadt, GER), EDC [1-ethyl-3-(3-dimethylaminopropyl) carbodiimide hydrochloride] (Merck, Darmstadt, GER), phosphate buffered saline (PBS) (Merck, Darmstadt, GER), Quant-iT™ PicoGreen^®^ dsDNA Assay Kit (Invitrogen, CA, USA), Sircol soluble collagen Assay Kit (Biocolor, County Antrim, UK), Blyscan Sulfated Glycosaminoglycan assay Kit, proteinase K (Merck, Darmstadt, GER), Dulbecco’s Phosphate Buffered Saline (DPBS) (Sigma-Aldrich, MO, USA), Dulbecco’s modified eagle’s medium (DMEM) (Biowest, Nuaillé, France), dimethyl sulfoxide (DMSO) (Merck, Darmstadt, GER), 3-(4,5-cimethylthiazol-2-yl)-2,5-diphenyltetrazolium bromide, MTT (Merck, Darmstadt, GER), live/dead viability/cytotoxicity assay kit (Thermo Fisher Scientific, MA, USA), ethidium homodimer-1 (EthD-1) (Biotium, Fremont, CA, USA), Tween 20 (Sigma-Aldrich, MO, USA), MES (J.T Baker, NJ, USA), Eosin (Merck, Darmstadt, GER), alcian blue (Merck, Darmstadt, GER), hematoxylin (Merck, Darmstadt, GER), Triton X-100 (Merck, Darmstadt, GER), xylene (Merck, Darmstadt, GER), eosin-phloxine. Antibiotics (Penicillin-Streptomycin;10,000 U/mL) (Thermo Fisher Scientific, MA, USA), collagenase (Sigma-Aldrich, MO, USA), calcein AM (Thermo Fisher Scientific, MA, USA), were used without further purification. The immunofluorescence staining reagents containing Hoechst 33,258 (Sigma-Aldrich, MO, USA), Dako Fluorescent Mounting Medium (Sigma-Aldrich, USA), anti-collagen II primary antibody (Sigma-Aldrich, MO, USA), anti-sox9 primary antibody (Sigma-Aldrich, MO, USA), anti-aggrecan primary antibody (Thermo Fisher Scientific, MA, USA), anti-MMP13 primary antibody (Sigma-Aldrich, MO, USA), anti-α-smooth muscle actin-Cy3 antibody (anti-α-SMA-Cy3) (Sigma-Aldrich, MO, USA), Goat anti-Mouse IgG (H + L) secondary antibody, and Alexa Fluor 488 (Thermo Fisher Scientific, Waltham, MA, USA) were employed.

### 3.2. Fabrication of 3D Uniform Porous Hydrogel Scaffold of N_2_/Gelatin with Opened Cells

We used a planar flow-focusing microfluidic device made of polydimethylsiloxane [11,12] to generate monodisperse bubbles that self-assemble into highly ordered flowing lattices collected into disc-shape reservoirs through tubing. Foam production was not exposed to ambient air directly, improving foam stability. The liquid solution contained 7% Type A gelatin and 1% Pluronic F127 surfactant in deionized water, and was pumped into the liquid inlet using a PhD 2000 syringe pump (Harvard Apparatus, Holliston, MA, USA). The pressure of nitrogen mixed with perfluorohexane was monitored with a Heise PM pressure gauge. The orifice where the air stream was focused by the liquid flow was 40 μm in diameter. The aqueous gelatin/surfactant solution was introduced into a designed dimethylsiloxane (PDMS) microfluidic device, nitrogen gas was introduced into the microchannel, and then bubbles were formed. The nitrogen pressure and the flow rate of the aqueous gelatin/surfactant solution could be adjusted to form a stable bubble size. Microbubbles were generated in a focusing flow at input liquid flow = 20 μL/min and air pressure = 15 and 25 psi for microbubble sizes of 100 μm and 160 μm, respectively [11,12]. The bubbles were collected into a special mold and kept at 4 °C to solidify. The mold was immersed in a cross-linking agent solution of 0.5% EDC [1-ethyl-3-(3-dimethylaminopropyl) carbodiimide hydrochloride] at 4 °C for 24 h, and a three-dimensional uniform cross-linked porous hydrogel scaffold of N_2_/gelatin with close microbubbles was formed. Furthermore, degassing was carried to build up a three-dimensional uniform cross-linked porous hydrogel scaffold with open cells [11,12].

### 3.3. Physical Properties of EDC-Crosslinked Gelatin-Based Hydrogel Scaffolds

Scanning electron microscopy and conjugated focus microscopy were used to confirm the pore size and microstructure of EDC-crosslinked gelatin-based hydrogel scaffolds for tissue engineering. Additionally, the gelatin-swelling ratio was detected.

The swelling rate of the gelatin-based hydrogel scaffold exhibited the culture medium’s ability to absorb the scaffold. The EDC-crosslinked gelatin-based hydrogel scaffold with a diameter of 0.8 cm and thickness of 0.15 cm was soaked in water. After confirming that all pores of the scaffold were filled with water, it was wiped with wet tissue paper to remove surface droplets, and weighted as Ws. The scaffold was freeze-dried to remove the moisture of scaffold and weighted as Wd. The swelling ratio (P) could be calculated by the following Formula (1):P = (Ws − Wd)/Wd × 100% (1)

The mechanical properties of the EDC-crosslinked gelatin-based hydrogel scaffold, such as tensile strength and elongation at break, were determined by TA.XT-plus Texture analyzer. The tensile test was then performed by pulling off the EDC-crosslinked gelatin-based hydrogel scaffold at pretest speed test, test speed, post test speed of 1, 1 and 10 mm/s using a texture analyzer (TA.XT-plus Texture analyzer, SMS, Surrey, UK). The net length between the jaws was almost constant for all films, 20 mm. The texture analyzer was run at auto force mode with a trigger force of 5 g. From stress–strain curves, two parameters were calculated: tensile strength was calculated as maximum stress and elongation at break where the film was torn.

In vitro degradation experiments of hydrogel scaffolds could be studied. The EDC-crosslinked gelatin-based hydrogel scaffolds were freeze-dried and weighted (W_0_), then placed in a culture flask containing 10 mL of DPBS (pH 7.4), degraded in a hybridization box at 37 °C, and replaced with DPBS every three days. The EDC-crosslinked gelatin-based hydrogel scaffold was taken out and freeze-dried after one week, two weeks, four weeks, and six weeks, and weighed as W_t_. The weight loss could be calculated as follows:Weight loss (%) = (W_0_ − W_t_)/W_0_ × 100%(2)

### 3.4. Isolation and Culture of Chondrocyte

To study the effect of the chondrocyte culture with designed scaffold, isolation and culture of rabbit articular chondrocytes were performed and the morphological characteristics of the rabbit chondrocytes were observed. The rabbit articular chondrocytes were isolated from the rabbit knee joint and disinfected with alcohol. The slices of cartilage were washed with sterilized phosphate buffered saline (PBS) and then immersed in 5% penicillin-streptomycin-neomycin for 15 min. After treatment, the slices of cartilage were washed with PBS, digested in 2% collagenase for 12 h and centrifuged to obtain a cell pellet. The cell pellet was then resuspended in Dulbecco’s modified eagle’s medium (DMEM) and seeded to a 10 cm culture dish at the cell density of 5 × 10^5^. Chondrocytes were cultured in DMEM, supplemented with 50 mg/mL L-ascorbic acid, 10% fetal bovine serum, and 1% antibiotics (Penicillin-Streptomycin;10,000 U/mL) in an incubator set at 5% CO_2_, 37 °C. Cells were passaged after 70~80% confluents were formed. The medium was changed every 2~3 days [44]. Furthermore, the resulting chondrocytes containing cytoskeleton-specific proteins such as collagen type II, Sox 9, Aggrecan, and MMP13 could be stained by indirect immunofluorescence staining. The samples stained by immunofluorescence staining could be prepared by culturing on chamber slides directly, then fixing, blocking, incubating, and imaging the resulting slides with chondrocytes.

### 3.5. The Feasibility of Constructing Cartilage Tissue with EDC-Crosslinked Gelatin-Based Hydrogel Scaffolds

Rabbit knee joint chondrocytes were seeded into EDC-crosslinked gelatin-based hydrogel scaffolds at a concentration of 1 × 10^6^ cell/scaffold. Chondrocytes of appropriate density were cultured with the EDC-crosslinked gelatin-based hydrogel scaffold for one to three weeks of static and dynamic cell culture at 37 °C and 5% CO_2_, and biochemical quantitative analysis was performed. The scaffolds with optimal hole sizes for chondrocyte growth were selected based on the results of static culture. Furthermore, rabbit knee joint chondrocytes were dynamically cultured with the selected scaffold in a tidal bioreactor (BelloCell1) (Cesco Bioengineering Co., Hsinchu, Taiwan) for four weeks. A disc of porous carriers 10 mm in diameter was used for cell immobilization. The carriers were pre-packed in the bioreactor with the working volumes of 500 mL used in this study (BelloCell1 500) and pre-sterilized by gamma irradiation (intensity = 25 kGY) [45].

The cell viability and cell number were observed by live/dead assay and MTT assay. The cell viability was assessed using a live/dead viability/cytotoxicity assay kit and ethidium homodimer-1 (EthD-1) and calcein AM were employed. After coculture, the hydrogel was rinsed with PBS buffer and stained with 500 μL live/dead assay as well. Images were observed using a fluorescence microscope (Olympus IX71, inverted microscope, Nagano, Japan). The live and dead cells were stained green or yellow, respectively. For MTT assay, at different time points, one sample was taken from each culture and treated with 240 μL of 3-(4,5-dimethylthiazolyl-2)-2,5-diphenyltetrazolium bromide (MTT, 0.5 mg/mL; Sigma) in Dulbecco’s modified eagle medium (DMEM) at 37 °C for 4 h in dark. Insoluble formazan crystals reduced from MTT were extracted with dimethyl sulfoxide (DMSO). The absorbance (optical density,OD) at 570 nm of the extractant was detected.

The cell proliferation status was evaluated by DNA concentration analysis. Furthermore, GAG and total collagen secretion were used to evaluate the degree of cell cartilage. The Quant-iTTM PicoGreen R dsDNA Reagent kit was used to detect the content of intracellular DNA. Quantitative analysis of deoxyribonucleic acid (DNA) was carried out by using fluorescence emission at 480 nm and 520 nm. The Blyscan Sulfated Glycosaminoglycan assay reagent combination was used to detect the content of soluble glycosaminoglycan (GAG) in chondrocyte-hydrogel tissue. Quantitative analysis of GAG was carried out by using Sepctrophotometer with an absorbance of 656 nm. A Sircol Soluble Collagen Assay Kit was used to measure the total collagen content of each sample. The samples were measured by using an immunofluorescence analyzer with a wavelength of 540 nm, then calculated with a standard calibration line.

### 3.6. Hematoxylin & Eosin Staining (H&E)

The tissue/scaffold samples were processed according to standard histology procedures, being fixed in formalin (10%), dehydrated through alcohol and xylene passages, and embedded in a block of paraffin wax for slicing by a microtome (cross section). After rehydration, the slides were first stained with hematoxylin solution for 8 min, then washed with water for 10 min, dripped with a few drops of 1% acid alcohol (1% HCl in 70% alcohol) for 30 s, washed in tap water for 1–5 min, and then washed with 0.2% ammonia water for 30 s. Then after 1 min, were again rinsed with running water for 5 min, then rinsed with 10 drops of 95% alcohol, eosin-phloxine for 30 s to 1 min. Finally, the samples were dehydrated. The nucleus was blue and the cytoplasm was pink or red.

### 3.7. Alcian Blue Staining

The tissue/scaffold samples were processed according to standard histology procedures, being fixed in formalin (10%), dehydrated through alcohol and xylene passages, and embedded in a block of paraffin wax for slicing by a microtome (cross section). After rehydration, the slides were first stained with alcian blue solution for 30 min, washed with running water for 2 min, stained with Nuclear fast red solution for 5 min, washed with running water for 1 min, and finally dehydrated. The nucleus was pink or red; the cytoplasm was light pink, and the weakly acidic sulfated mucosubstances, hyaluronic acid, and sialomucins were dark blue.

### 3.8. Immunofluorescence Stain

In this experiment, cytoskeleton-specific proteins such as collagen type II, Sox 9, Aggrecan, and MMP13 were stained by indirect immunofluorescence staining. Cultured cells grown in scaffolds after 7 and 21 days were fixed with cold methanol for 5–10 min, proteinase K was added, then pre-warmed to 37 °C for 15 min to reduce the antigen, and rinsed four times with 0.05% Tween 20/phosphate-buffered saline (PBS), 0.1% Triton X-100/PBS was added for 5 min, then blocking was perfomed, 5% fetal bovine serum was used for 45 min, then they were washed with 0.05% Tween 20/PBS, the primary antibody was selected for one hour, and 0.05% Tween 20/Rinse with PBS, using the dye-labeled specific protein as fluorescein-conjugated secondary antibodies for one hour, rinsed with 0.05% Tween 20/PBS, stained nuclei with Hoechst 33,258 (1:10,000) for 5 min, rinsed with 0.05% Tween 20/PBS, and added anti-α-smooth muscle actin-Cy3 (anti-α-SMA-Cy3) (1:1000) to stain the cytoplasm for 10–15 min, rinsed with 0.05% Tween 20/PBS, and mounted with Dako Fluorescent Mounting Medium when it was dry. After drying in the shade, the four sides were sealed for preservation.

### 3.9. Statistical Analysis

The obtained results were expressed as mean values ± standard error of mean (SEM) (n = 3). The Shapiro–Wilk W test was used to verify the distribution of all variables. For the normally distributed variables, comparisons among extraction protocols were made with one-way analysis of variance (ANOVA) with post hoc Tukey multiple comparison tests. In the case of non-normally distributed variables, the Kruskal–Wallis test was used. GraphPad Prism version 8.1.0 was used for the analyses. Statistical differences were considered significant at *p* < 0.05.

### 3.10. Measurements

The morphology of the chondrocytes grown on the EDC-crosslinked gelatin-based hydrogel scaffold was studied by scanning electron microscopy (SEM) (S3400N, Hitachi, Tokyo, JP), confocal microscope (Leica TCS SP2), and Microscope (AE2000, Motic, Schertz, TX, USA). A UV/vis spectrophotometer (GENESYS 10S, Thermo Fisher Scientific, MA, USA) was employed for quantitative analysis of GAG. An immunofluorescence analyzer (CTK Biotech, Poway, CA, USA) was employed for the study of the immunofluorescence staining.

## 4. Conclusions

Using microfluidic technology, three-dimensional gelatin-based hydrogel scaffolds with interconnected uniform open cells could be successfully prepared for application with tissue engineering. The three-dimensional gelatin-based hydrogel scaffolds could provide good rabbit chondrocyte cell growth. The static culture results showed that the three-dimensional pores in a porous scaffold could promote chondrocyte cell growth. The chondrocytes crawl to the center of the scaffold for a long time, although the gelatin-based hydrogel scaffolds had a three-dimensional porous structure with highly organized and interconnected pores. A tidal bioreactor was employed to improve the exchange rate of the culture medium and gas in the gelatin-based hydrogel scaffolds, and to simulate the process in vivo. From the dynamic culture results, the gelatin-based hydrogel scaffold with uniform open cells exhibited good cell growth promotion performance. The gelatin-based hydrogel scaffold encouraged chondrocyte proliferation, promoting the expression of collagen type II, aggrecan, and sox9, while retaining structural stability and cartilage durability after dynamic compression, as well as promoting cartilage repair.

## Figures and Tables

**Figure 1 ijms-23-08449-f001:**
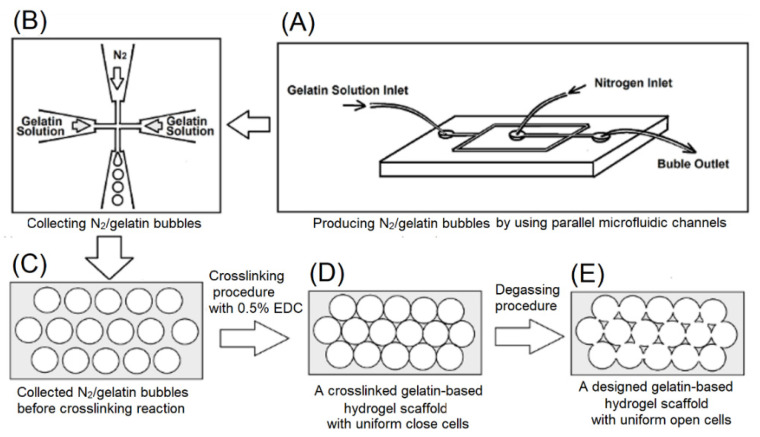
Schematic illustrations for the fabrication process of highly organized EDC-crosslinked porous gelatin-based hydrogel scaffolds, (**A**) producing N_2_/gelatin bubbles by using parallel microfluidic channels, (**B**) collecting N_2_/gelatin bubbles, (**C**) collected N_2_/gelatin bubbles before crosslinking reaction, (**D**) a crosslinked gelatin-based hydrogel scaffold with uniform close cells, and (**E**) a designed gelatin-based hydrogel scaffold with uniform open cells.

**Figure 2 ijms-23-08449-f002:**
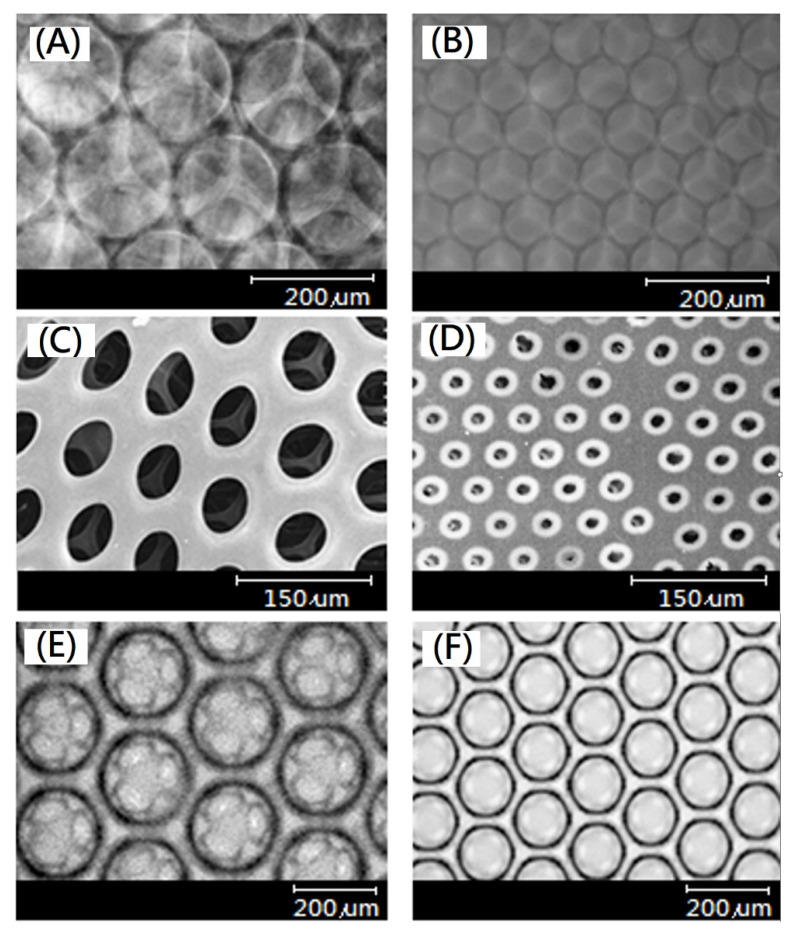
Microscope results of EDC-crosslinked gelatin-based hydrogel scaffolds having (**A**) pores with diameter of 160 μm and (**B**) pores with diameter of 100 μm, scanning electron microscope of (**C**) pores with diameter of 160 μm and (**D**) pores with diameter of 100 μm, and conjugate focus fluorescence microscopy of (**E**) pores with diameter of 160 μm and (**F**) pores with diameter of 100 μm.

**Figure 3 ijms-23-08449-f003:**
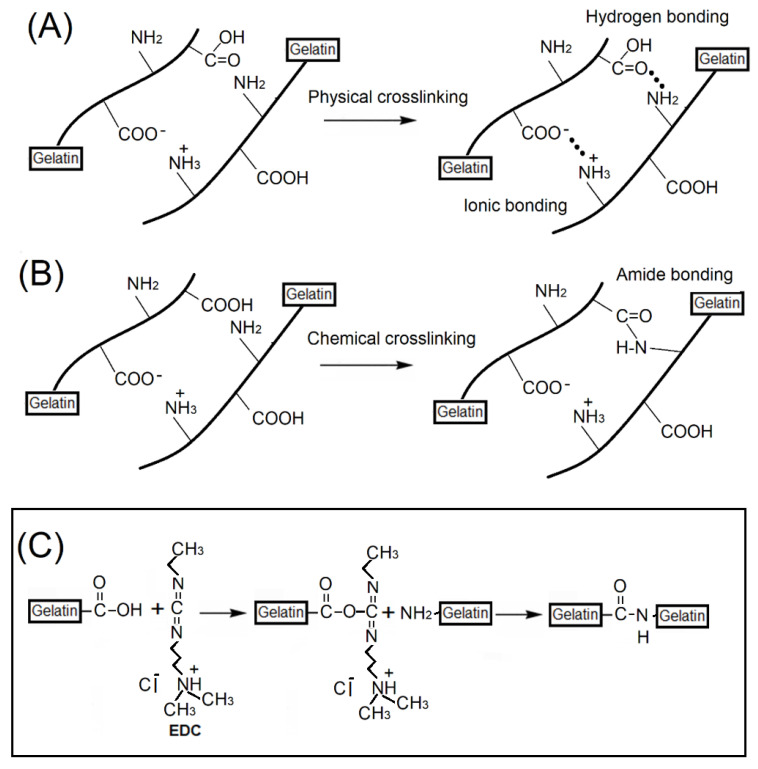
Schematic representing the possible interactions within gelatin molecules, formation of hydrogen bonds during swelling and amide bonds via (**A**) physical and (**B**) chemical crosslinking behaviors, in which a crosslinking agent of EDC would be used (**C**).

**Figure 4 ijms-23-08449-f004:**
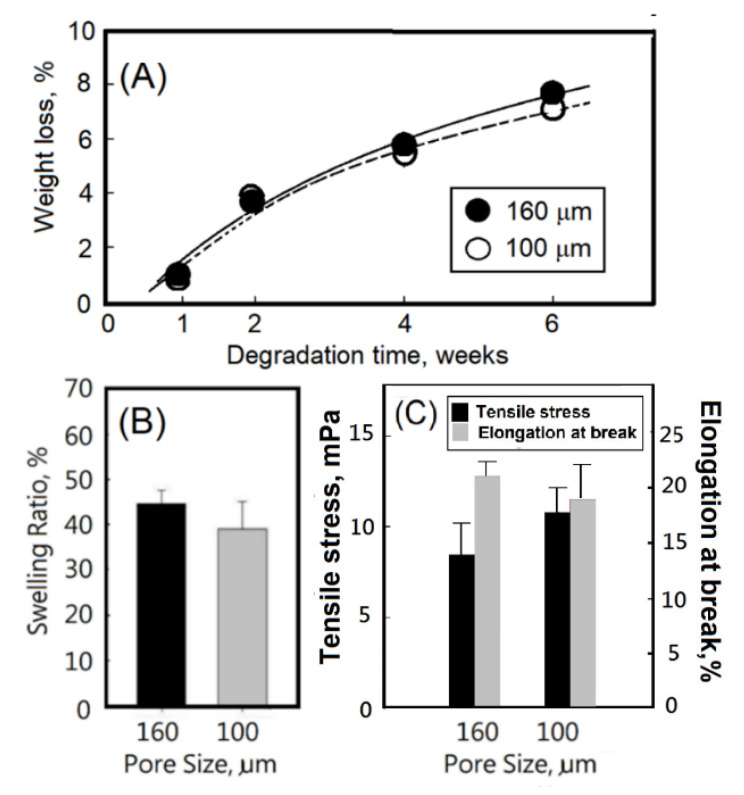
(**A**) Degradation rate, (**B**) swelling ratio, and (**C**) tensile strength and elongation at break of EDC-crosslinked gelatin-based hydrogel scaffolds with different pore sizes (160 and 100 μm). Results were expressed as the means ± SEM (n = 3).

**Figure 5 ijms-23-08449-f005:**
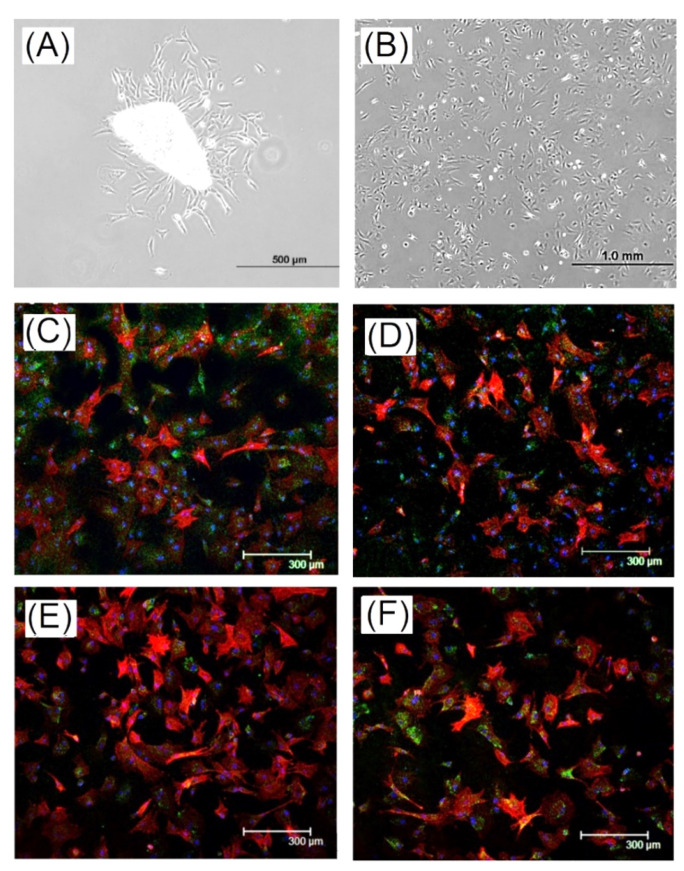
O.D. images of (**A**) culture of rabbit knee articular cartilage fragment (chondrocyte) and (**B**) primary chondrocytes (P0) (40 x) and immunofluorescence staining results of chondrocytes, green-specific protein, (**C**) type II collagen, (**D**) MMP13, (**E**) aggrecan, and (**F**) sox9; blue-nucleus and red-cytoskeleton.

**Figure 6 ijms-23-08449-f006:**
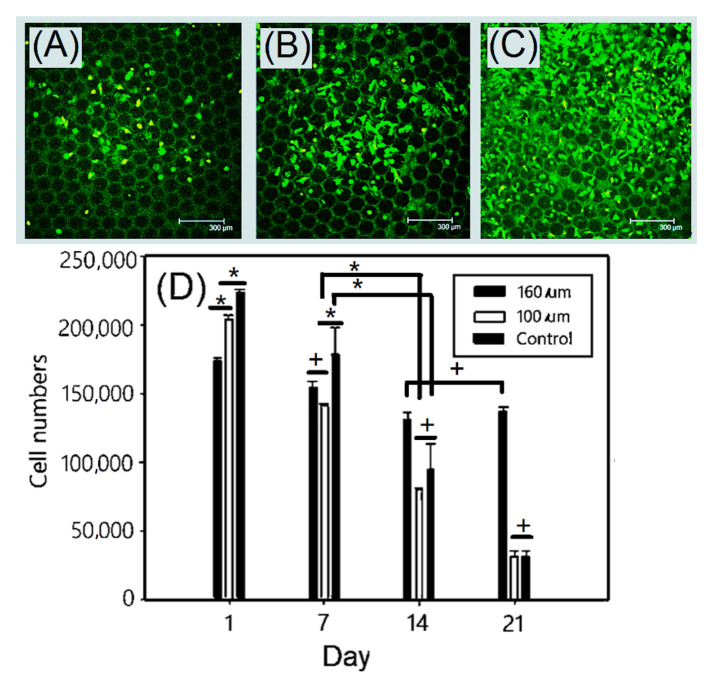
Live/dead cytotoxicity viability assay of chondrocytes on the gelatin-based hydrogel scaffold after (**A**) 1, (**B**) 4, and (**C**) 7 days (Scale bar = 300 μm) and (**D**) MTT assay results of chondrocytes on the gelatin-based hydrogel scaffold. Green: live cells, red: dead cells, orange or yellow: overlap of live and dead cells. Values with different superscripts were significantly different (*p* < 0.05); *, statistical significance between the groups (*p* < 0.05); +, nonsignificance within the group (*p* > 0.05).

**Figure 7 ijms-23-08449-f007:**
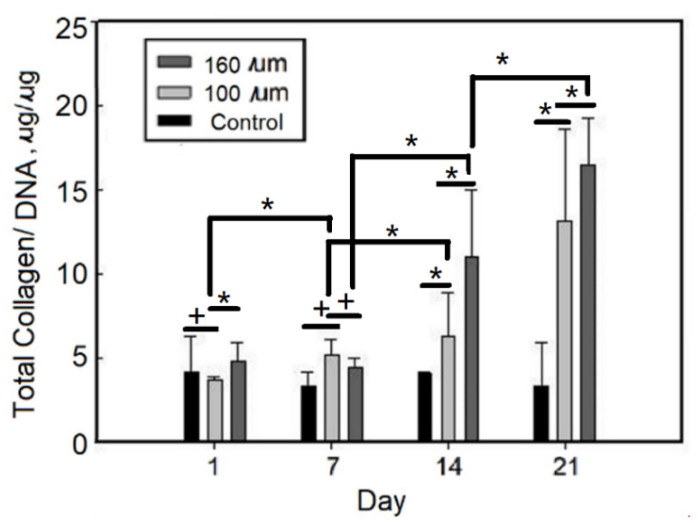
Total collagen/DNA of static culturing chondrocytes on EDC-crosslinked gelatin-based hydrogel scaffolds. Values with different superscripts were significantly different (*p* < 0.05); *, statistical significance between the groups (*p* < 0.05); +, nonsignificance within the group (*p* > 0.05).

**Figure 8 ijms-23-08449-f008:**
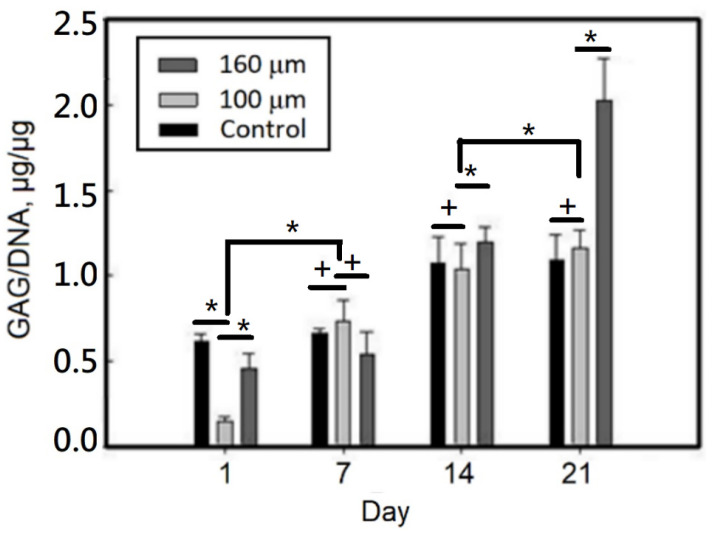
GAG/DNA of static culturing chondrocytes on gelatin-based hydrogel scaffolds. Values with different superscripts were significantly different (*p* < 0.05); *, statistical significance between the groups (*p* < 0.05); +, nonsignificance within the group (*p* > 0.05); n = 6.

**Figure 9 ijms-23-08449-f009:**
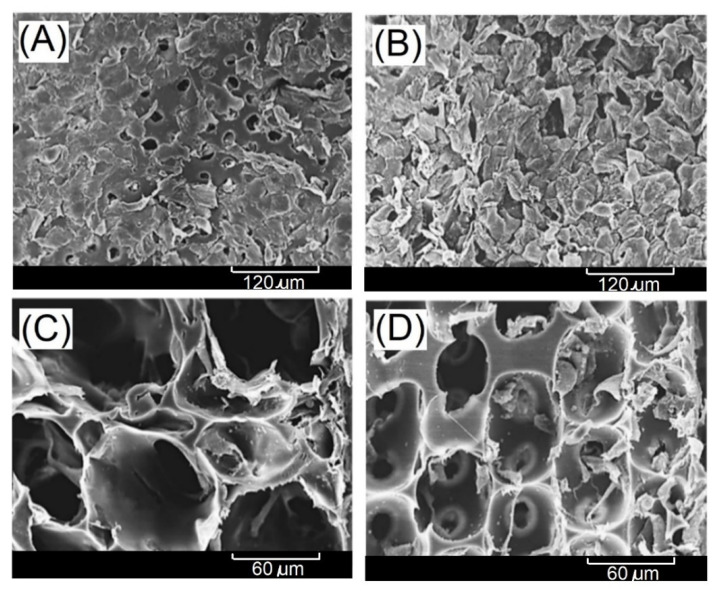
Scanning electron microscope analysis of static culturing chondrocytes on EDC-crosslinked gelatin-based hydrogel scaffolds, (**A**) after 7 days (100 μm) (surface, scale bar = 120 μm), (**B**) after 7 days (160 μm) (surface, scale bar = 120 μm), (**C**) after 14 days (100 μm) (cross section, scale bar = 60 μm), and (**D**) after 14 days (160 μm) (cross section, scale bar = 60 μm).

**Figure 10 ijms-23-08449-f010:**
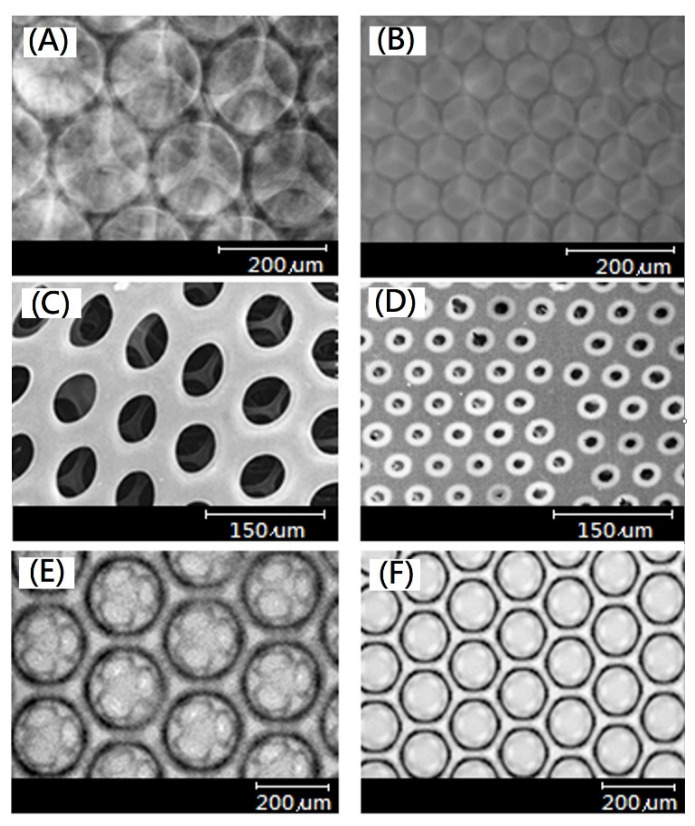
SEM morphology of chondrocyte grown on EDC-crosslinked gelatin-based hydrogel scaffold after dynamic culture, wherein (**A**) surface (scale bar = 200 μm) and (**B**) cross section (scale bar = 50 μm) after 7 days, (**C**) surface (scale bar = 2 00 μm) and (**D**) cross section (scale bar = 50 μm) for 14 days, and (**E**) surface (scale bar = 200 μm) and (**F**) cross section (scale bar = 50 μm) for 21 days.

**Figure 11 ijms-23-08449-f011:**
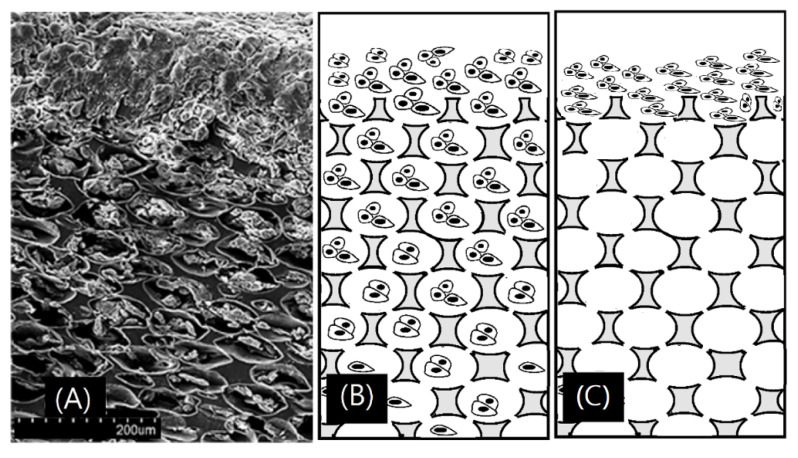
(**A**) SEM morphology of chondrocyte grown on EDC-crosslinked gelatin-based hydrogel scaffold after dynamic culture for 28 days(cross section) (scale bar = 200 μm), (**B**) Schematic drawing of chondrocyte growth on EDC-crosslinked gelatin-based hydrogel scaffold after dynamic culture (cross section), and (**C**) Schematic drawing of chondrocyte growth on EDC-crosslinked gelatin-based hydrogel scaffold after static culture (cross section).

**Figure 12 ijms-23-08449-f012:**
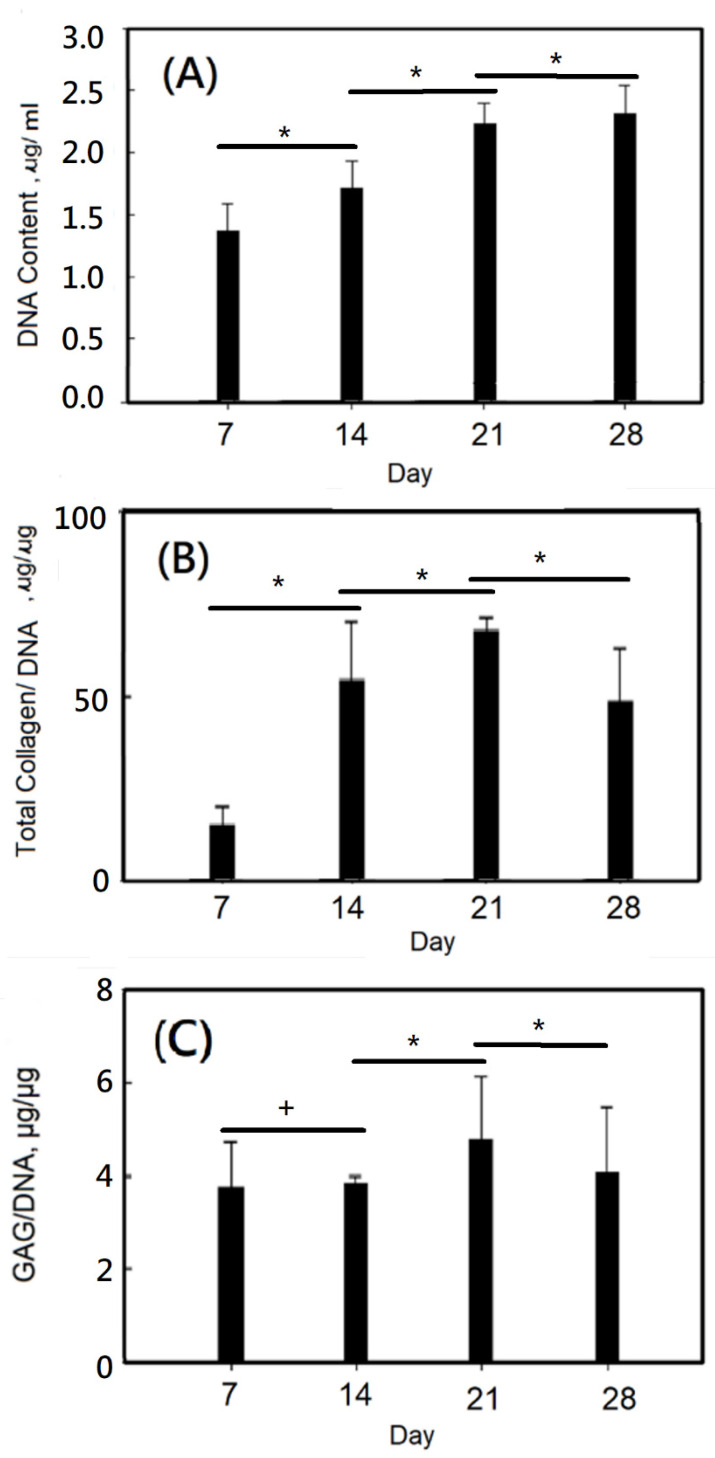
(**A**) DNA content of dynamic culturing chondrocytes on gelatin-based hydrogel scaffolds, (**B**) total collagen/DNA of dynamic culturing chondrocytes on gelatin-based hydrogel scaffolds, and (**C**) GAG/DNA of dynamic culturing chondrocytes on gelatin-based hydrogel scaffolds. Values with different superscripts were significantly different (*p* < 0.05); *, statistical significance between the groups (*p* < 0.05); +, nonsignificance within the group (*p* > 0.05); n = 6.

**Figure 13 ijms-23-08449-f013:**
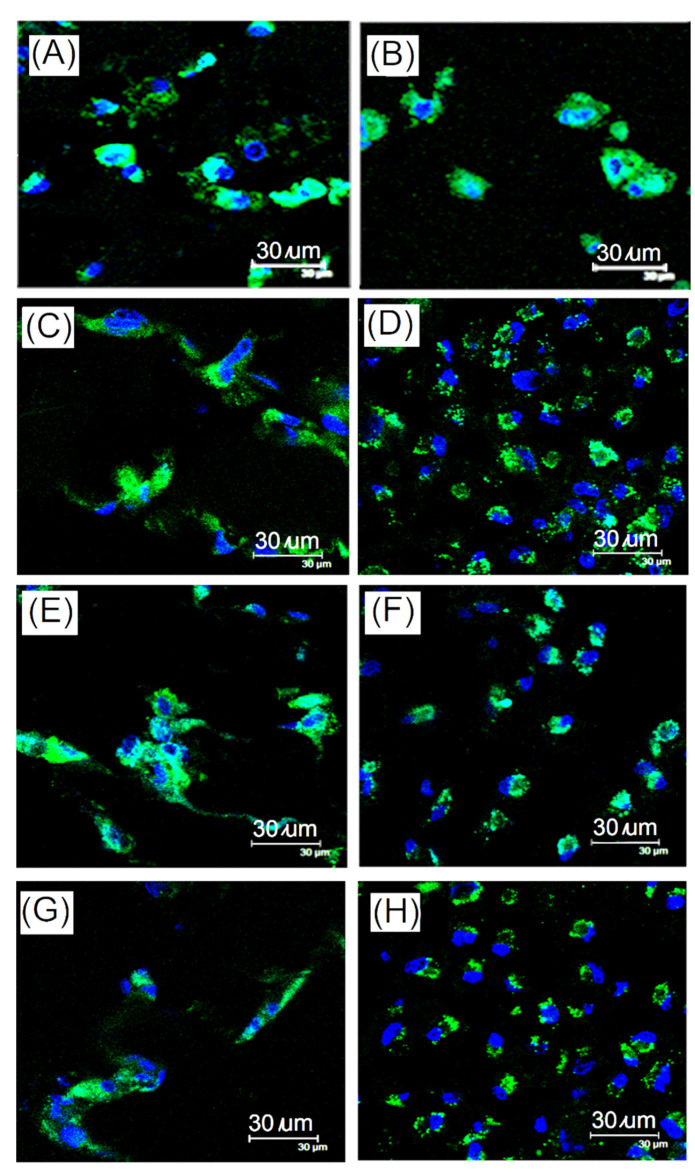
Immunofluorescence staining of expression from chondrocytes grown on EDC-crosslinked gelatin-based hydrogel scaffold in dynamic culture system for collagen type II (**A**) after 21 days and (**B**) after 28 days (blue-Nuclei and green-collagen type II); for MMP13 (**C**) after 21 days and (**D**) after 28 days (blue-nucleus and green-MMP13); sox9 (**E**) after 21 days and (**F**) after 28 days (blue-nucleus and green-sox9); and aggrecan (**G**) after 21 days and (**H**) after 28 days(blue-nucleus and green-aggrecan) (scale bar, 30 μm).

**Figure 14 ijms-23-08449-f014:**
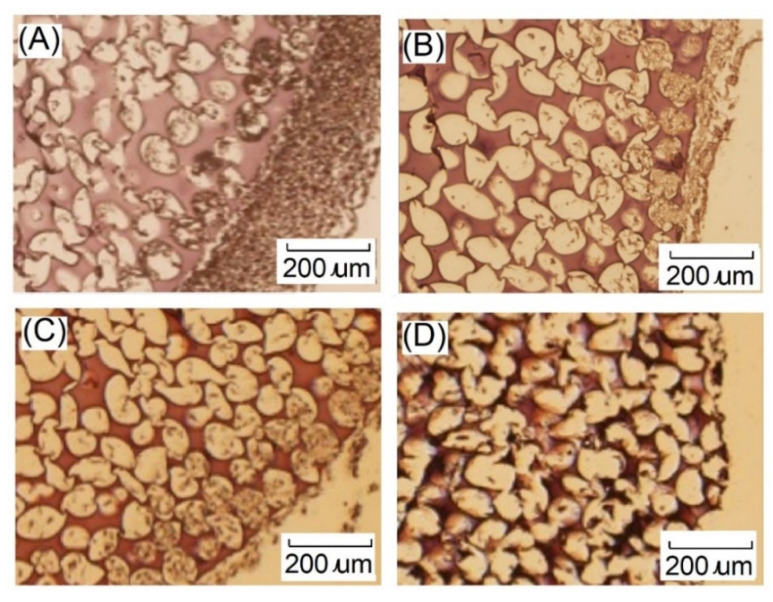
Images of hematoxylin and eosin staining showing growth of chondrocytes on gelatin-based hydrogel scaffolds after (**A**) 7 days, (**B**) 14 days, (**C**) 21 days, and (**D**) 28 days of dynamic culture. Dark blue-nucleus, light red-cytoplasm, intercellular substance, red-hydrogel scaffold. Scale bar represents 200 μm. .

**Figure 15 ijms-23-08449-f015:**
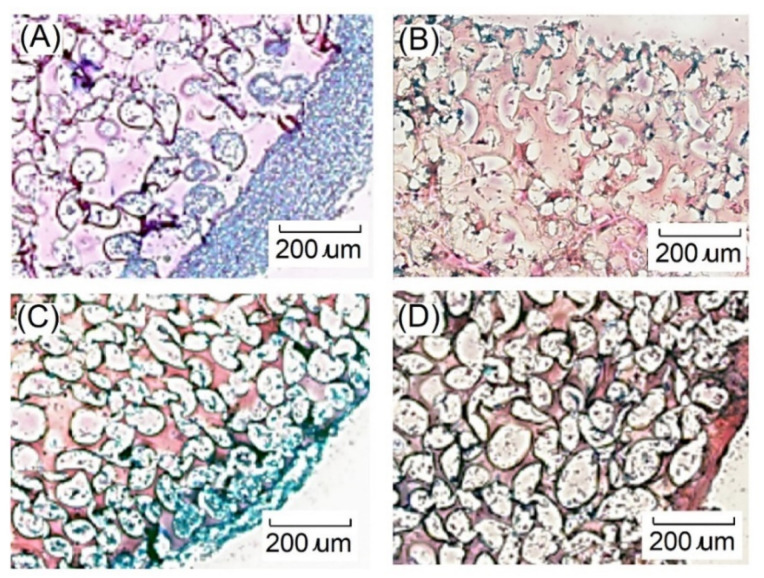
Images of alcian blue staining showing growth of chondrocytes on gelatin-based hydrogel scaffolds after (**A**)7 days, (**B**) 14 days, (**C**) 21 days, and (**D**) 28 days of dynamic culture. Blue-glycosaminoglycan, red-nucleus, dark red-hydrogel scaffold. Scale bar represents 200 μm.

## Data Availability

The original contributions presented in the study are included in the article, further inquiries can be directed to the corresponding author.

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
