# Peer review of "Highly Organized Porous Gelatin-Based Scaffold by Microfluidic 3D-Foaming Technology and Dynamic Culture for Cartilage Tissue Engineering"

_ijms, 2022, doi:10.3390/ijms23158449_

Round 1

Reviewer 1 Report

The paper deals with an interesting subject, which is the design, development and characterization of a gelatin-based hydrogel scaffold with highly uniform pore size and biocompatibility for cartilage tissue engineering. The authors used microfluidic 3D-foaming technology for scaffold production. Furthermore, they deeply analysed the scaffolds form a morphological, mechanical and biological point of view.

The paper is of clear interest for the readers of the journal as well as all the regenerative medicine community.

Nevertheless, the paper is not suitable for publication in the present form and some minor changes should be properly addressed by the authors to improve the clarity and the understanding.

1.  The authors should improve the English, in order to improve the readability of the mannuscript.

2.     The authors should improve the “Introduction” section. In particular, it results confusing and does not flow properly. The authors should report other works highlighting the advantages of using the microfluidic route in the production of three-dimensional scaffolds. As for example, did the authors evaluate the possibility of using 3D bioprinting for the same objective?

3.  On page 8, “Comparing the swelling 288 ratio of scaffolds with different pore sizes, the gelatin-based hydrogel scaffolds with uni-289 form pores of 160μm diameter showed relative high swelling ratio as shown in Figure 290 4(B). Did the authors compare their results with the literature?

4.   On page 8, “Also, the elasticity of gelatin-based 292 hydrogel scaffolds was detected. Comparing the gelatin-based hydrogel scaffold with different pore sizes, the larger the pore size and the greater the elasticity of the scaffold as 294 shown in Figure 4(C).” Did the authors compare their results with the literature?

5.   In general, the authors should compare their results with the literature.

6.    I would suggest also to reorganize the figures in panels. In such a way, the final number of figures would be reduced with the consequence to have a more clear and nice paper.

Minor changes:

1.     In the abstract, the authors should replace “ccurture” with “culture”

2.     In the abstract, the authors should also report the main achievements.

3.    The authors should describe the acronyms the first time they appear in the main text.

4.   In “Materials and Methods”, the authors did not report a section on the used Chemicals (i.e. Company and Nationality). Which gelatin did they employ? A or B? Furthermore, some methods are missing (i.e. SEM). The authors should enrich this section highlighting more information. With regard to the mechanical analysis, did the authors use a standard? How many samples did they test? The authors should also report the size of the samples.

5.  Section 2.4, the authors should report the cell density. This information is reported in the “Results” section.

6.     On page 11, “As shown in Figure 6, live cells were stained with green fluorescence, and dead cells were 347 stained with yellow fluorescence”. The difference between the two colors is not clear for the reader. For this reason, the statement reported above is not immediate.

7.   On page 11, this sentence: “Figure 8. The 160μm hydrogel scaffold had slightly higher secretion than the 100μm pore size total collagen/DNA” should be remove.

8.    Figures are not correctly cited in the main text. The authors should double check the number of figures and figure captions.

Author Response

  1. The authors should improve the English, in order to improve the readability of the manuscript.

As reviewer 1’s suggestion, some descriptions had been corrected and modified to improve the readability of the revised manuscript.

  1. The authors should improve the “Introduction” section. In particular, it results confusing and does not flow properly. The authors should report other works highlighting the advantages of using the microfluidic route in the production of three-dimensional scaffolds. As for example, did the authors evaluate the possibility of using 3D bioprinting for the same objective?

As reviewer 1’s suggestion, other works highlighting the advantages of using the microfluidic route in the production of three-dimensional scaffolds had been added in the revised manuscript. The porous scaffold with a three-dimensional microstructure was considered for applications of cartilage tissue engineering. However, the porous scaffold with uniform three-dimensional holes and high interconnectivity was difficult to prepare. For these reasons, several techniques such as bioprinting procedures[9] and microfluidic procedures were studied[10-12]. Particularly, some conventional microfluidic systems were designed and employed to fabricate scaffolds efficiently with a desirable homogenous porosity, uniform spatial structure, interconnectivity, and potential mechanical properties[10-12]. For appli-cation purposes, these conventional microfluidic techniques had been applied to fabricate new materials that encapsulate target cells in small spherical pore and foam which would be a powerful method for tissue engineering [10-12].

  1. On page 8, “Comparing the swelling 288 ratio of scaffolds with different pore sizes, the gelatin-based hydrogel scaffolds with uni-289 form pores of 160μm diameter showed relative high swelling ratio as shown in Figure 290 4(B). Did the authors compare their results with the literature?

As reviewer 1’s suggestion, the results had been compared with literature. The description had also be modified.

  1. On page 8, “Also, the elasticity of gelatin-based 292 hydrogel scaffolds was detected. Comparing the gelatin-based hydrogel scaffold with different pore sizes, the larger the pore size and the greater the elasticity of the scaffold as 294 shown in Figure 4(C).” Did the authors compare their results with the literature?

As reviewer 1’s suggestion, the results had been deleted to avoid confusion.

  1. In general, the authors should compare their results with the literature.

As reviewer 1’s suggestion, some results had been compared with the literature.

  1. I would suggest also to reorganize the figures in panels. In such a way, the final number of figures would be reduced with the consequence to have a more clear and nice paper.

As reviewer 1’s suggestion, the figures in panels had been reorganized and the final number of figures had been reduced.

Minor changes:

  1. In the abstract, the authors should replace “ccurture” with “culture”

As reviewer 1’s suggestion, the “ccurture” had been replaced with “culture”.

  1. In the abstract, the authors should also report the main achievements.

As reviewer 1’s suggestion, the main achievements had been reported in the abstract of revised manuscript.

  1. The authors should describe the acronyms the first time they appear in the main text.

As reviewer 1’s suggestion, the acronyms had been described at the first time they appear in the main text.

  1. In “Materials and Methods”, the authors did not report a section on the used Chemicals (i.e. Company and Nationality). Which gelatin did they employ? A or B? Furthermore, some methods are missing (i.e. SEM). The authors should enrich this section highlighting more information. With regard to the mechanical analysis, did the authors use a standard? How many samples did they test? The authors should also report the size of the samples.

As reviewer 1’s suggestion, the chemicals with Company and Nationality had been corrected. Type A gelatin (Sigma-Aldrich, St. Louis, MO, USA) and 1% Pluronic F127(Invitrogen, UK) had been described in the revised manuscript.  Some methods had been described in the revised manuscript as reviewers’ suggestions such as SEM, MTT assay, LIVE/DEAD assay, condition of dynamic culture, etc. The results of mechanical analysis had been deleted to avoid confusion. Size of the samples had been reported as a statement of  ” The gelatin-based hydrogel scaffold with diameter of 0.8cm and thickness of 0.15 cm.”.

  1. Section 2.4, the authors should report the cell density. This information is reported in the “Results” section.

As reviewer 1’s suggestion, “cell density” had been reported in Section 2.4 of revised manuscript.

  1. On page 11, “As shown in Figure 6, live cells were stained with green fluorescence, and dead cells were 347 stained with yellow fluorescence”. The difference between the two colors is not clear for the reader. For this reason, the statement reported above is not immediate.

As reviewer 1’s suggestion, the difference between the two colors had been enhanced to make the results clearly.

  1. On page 11, this sentence: “Figure 8. The 160μm hydrogel scaffold had slightly higher secretion than the 100μm pore size total collagen/DNA” should be remove.

As reviewer 1’s suggestion, “Figure 8. The 160μm hydrogel scaffold had slightly higher secretion than the 100μm pore size total collagen/DNA” had been removed in the revised manuscript.

  1. Figures are not correctly cited in the main text. The authors should double check the number of figures and figure captions.

 As reviewer 1’s suggestion, figures and figure captions had been checked and corrected with corresponding discussions in the revised manuscript.

Reviewer 2 Report

This manuscript described an approach for the formation of a gelatin-based scaffold for cartilage cell culture.  The experiments were not carefully designed and the results are not convincing.  There are some ethical issues regarding how the rabbit cartilage tissue was isolated.  There is no protocol that was approved.  

Author Response

 As reviewer 2’s suggestion, statement of “Animal Ethics” had been added in the revised manuscript.  The information was supplied relating to ethical approvals: Animal study was approved by Institutional Animal Care and Use Committee of Master Laboratory Co. Ltd, Taiwan(IACUC Approval No.:20201210). Results and Figures had been checked and corrected with corresponding discussions in the revised manuscript.

Reviewer 3 Report

The manuscript reports the fabrication of gelatin hydrogels with controlled pore size and interconnectivity by using microfluidic 3D-foaming technology. The technology was used to create chemically crosslinked hydrogels of different diameters, which were evaluated regarding the swelling and mechanical properties, as well as cellular response. Please see my specific comments regarding key issues that need improvement before publishing.

1. A similar approach for porous hydrogel fabrication has been previously reported in refs 11 and 12, as stated in the manuscript. In light of this, what are the distinctive features of the manuscript? The main novelty and contribution of the paper to the field need to be clarified.

2. In lines 41-43, hydrogels should be focused on cartilage application, which is the focus of the manuscript. Similarly, text in lines 49-52 is unclear; authors should clarify the meaning of such sentence. 

3. Lines 54-55: author state that it’s difficult to prepare scaffolds with uniform porosity and pore interconnectivity. Despite this is true when conventional processes are used for hydrogel fabrication, the use of additive manufacturing allows the fabrication of hydrogels with controlled porosity and pore size. Indeed, there are several works focusing on hydrogel bioprinting or combination between bioprinting technologies to create complex scaffolds for cartilage tissue engineering. Thus, this should be discussed/clarified in the manuscript.

4. In section 3.2, authors focus on the bulk and surface erosion of polymers. However, there is no research data supporting the mechanism of degradation of gelatin hydrogels.

5. Fig. 4: authors need to clarify what “Elasticity” means and how it was calculated. In addition, it would be more compelling to provide data about the porosity of hydrogels and mechanical properties such as elastic modulus. Fig. 5: data could be moved to SI as this is well-established in the field. Further, the color code is missing in the figure legend. Data in Figs. 7 and 8 is poorly discussed in the text. 

6. Authors should clarify the rational for selecting hydrogel scaffolds with pore size of 160μm for dynamic culture.

7. Figs 10-15: some of the claims made by the authors regarding these figures are not fully supported by research data. In detail:

- “gelatin-based hydrogel scaffold with pore size of 160μm diameter had a relative high 395 amount of extracellular matrix secretion”. How do authors quantify the amount of ECM based on SEM images? It would be more compelling to perform immunofluorescence staining and quantitative analysis of the data.

- Line 441: “type II collagen gradually increased after 28 days.” How did the authors quantify the increase/decrease in ECM? From images it seems that there are differences in cell number, which could impact ECM deposition. How many images and independent experiments were considered to draw such conclusions? This is unclear.

- Line 444: “After dynamic culture, the number of cells increased, and type II collagen increased”. Despite this statement, it is unclear whether cell proliferation is improved in dynamic culture compared to static culture as well as whether expression of specific makers is influenced by dynamic culture. Authors should address these points adding experimental data quantifying cell proliferation and ECM deposition.

- Line 477: “The dynamic reactor would improve cell proliferation dramatically”. Data is not provided.

8. Statistical analysis in missing in the manuscript.

9. The main conclusion(s) of the manuscript should be included in the abstract.

10. Experimental methods: more details on the dynamic culture and parameters hydrogel stimulation should be added. 

11. Figures: confirm that all figures appear in proper locations in the text.

12. The manuscript has several English spelling issues and requires revision by a native speaker. Some of the issues are detailed bellow.

- Line 16: “bubbles were corrected.” What this it means? 

- Line 21: “decomposition rate.” Authors wanted to say degradation?

- Line 24: “comfirm the active.” 

- Line 26: “ccurture.”

- Line 59: “Gelatin was a mixture”

- Line 83: “cartilage tissue engineering was important.” Important for what? 

- Line 131: “hydrogel scaffold s”

- Line 140: “cartilage was harvested from the rabbit knee joint was disinfected”

- Line 457: “chondrocytes were planted in gelatin”

- Line 470: “staining 470 methods in section staining”

- Line 473: “From the results in Figure 14, after seven days of static cell culture, which represents 473 cells’ localization in scaffolds”

- Line 497: “Using the of microfluidic technology”

Author Response

  1. A similar approach for porous hydrogel fabrication has been previously reported in refs 11 and 12, as stated in the manuscript. In light of this, what are the distinctive features of the manuscript? The main novelty and contribution of the paper to the field need to be clarified.

As reviewer 3’s suggestion, the main novelty and contribution of the paper(refs 11 and 12) to the field had been described in the revised manuscript. The porous scaffold with a three-dimensional microstructure was considered for applications of cartilage tissue engineering. However, the porous scaffold with uniform three-dimensional holes and high interconnectivity was difficult to prepare. For these reasons, several techniques such as bioprinting procedures[9] and microfluidic procedures were studied[10-12]. Particularly, some conventional microfluidic systems were designed and employed to fabricate scaffolds efficiently with a desirable homogenous porosity, uniform spatial structure, interconnectivity, and potential mechanical properties[10-12]. For appli-cation purposes, these conventional microfluidic techniques had been applied to fabricate new materials that encapsulate target cells in small spherical pore and foam which would be a powerful method for tissue engineering [10-12].

  1. In lines 41-43, hydrogels should be focused on cartilage application, which is the focus of the manuscript. Similarly, text in lines 49-52 is unclear; authors should clarify the meaning of such sentence. 

As reviewer 3’s suggestion, text in lines 49-52 had been modified to make it clear. The cartilage application had been added in the revised manuscript(lines 43-45 & 49-53).

  1. Lines 54-55: author state that it’s difficult to prepare scaffolds with uniform porosity and pore interconnectivity. Despite this is true when conventional processes are used for hydrogel fabrication, the use of additive manufacturing allows the fabrication of hydrogels with controlled porosity and pore size. Indeed, there are several works focusing on hydrogel bioprinting or combination between bioprinting technologies to create complex scaffolds for cartilage tissue engineering. Thus, this should be discussed/clarified in the manuscript.

As reviewer 3’s suggestion, some works focusing on hydrogel bioprinting or combination between bioprinting technologies had been added in the revised manuscript. To enhance stability and mechanical properties, the gelatin could be covalently cross-linked by using chemical crosslinking agents such as formaldehyde, glutaraldehyde, par-aformaldehyde, 1-ethyl-3-(3-dimethylaminopropyl)carbodiimide hydrochloride(EDC), and genipin[19-21]. Also, enzymatic crosslinking procedure for crosslinking reatgion of gelatin studied by using enzymes such as tyrosinase and transglutaminase[22,23]. Other physical crosslinking methods, such as plasma treatment, often result in low crosslinking extent of gelatin macromolecules because crosslinking occurs only at the surface of the material[24]. Photocrosslinked gelatin-based hydrogels usually present short gelation time and are chemically stable and mechanically strong, but photo-initiators required for the photopolymerization reaction might lead to cell death[25]. Furthermore, gelatin can also be cross-linked using microbial transglutaminase (MTG), an FDA-approved enzyme that covalently bonds the glutamine and lysine groups between gelatin polymers[26].

  1. In section 3.2, authors focus on the bulk and surface erosion of polymers. However, there is no research data supporting the mechanism of degradation of gelatin hydrogels.

 As reviewer 3’s suggestion, the statement had been modified.

  1. Fig. 4: authors need to clarify what “Elasticity” means and how it was calculated. In addition, it would be more compelling to provide data about the porosity of hydrogels and mechanical properties such as elastic modulus. Fig. 5: data could be moved to SI as this is well-established in the field. Further, the color code is missing in the figure legend. Data in Figs. 7 and 8 is poorly discussed in the text. 

As reviewer 3’s suggestion, results of “Elasticity” had been deleted to avoid confusion. Fig.5 had been considered to move to S1 The discussion of Figs. 7 and 8 had been modified.

  1. Authors should clarify the rational for selecting hydrogel scaffolds with pore size of 160μm for dynamic culture.

As reviewer 3’s suggestion, the rational for selecting hydrogel scaffolds with pore size of 160μm for dynamic culture had been clarity. Some results had been described and modified to support the rational for selecting hydrogel scaffolds with pore size of 160μm.  Most importantly, the chondrocytes-covered gelatin-based hydrogel scaffolds with relatively large pore sizes of 160μm might provide relative high interconnectivity for culture.  

  1. Figs 10-15: some of the claims made by the authors regarding these figures are not fully supported by research data. In detail:

- “gelatin-based hydrogel scaffold with pore size of 160μm diameter had a relative high 395 amount of extracellular matrix secretion”. How do authors quantify the amount of ECM based on SEM images? It would be more compelling to perform immunofluorescence staining and quantitative analysis of the data.

As reviewer 3’s suggestion, the amounts of ECM were quantified by total collagen/DNA secretion and the amount of GAG/DNA which had been described in the revised manuscipt. Figs 10-15 had been checked and corrected with corresponding discussions.

- Line 441: “type II collagen gradually increased after 28 days.” How did the authors quantify the increase/decrease in ECM? From images it seems that there are differences in cell number, which could impact ECM deposition. How many images and independent experiments were considered to draw such conclusions? This is unclear.

As reviewer 3’s suggestion, Figures had been checked and corrected with corresponding discussions in the revised manuscript.

- Line 444: “After dynamic culture, the number of cells increased, and type II collagen increased”. Despite this statement, it is unclear whether cell proliferation is improved in dynamic culture compared to static culture as well as whether expression of specific makers is influenced by dynamic culture. Authors should address these points adding experimental data quantifying cell proliferation and ECM deposition.

As reviewer 3’s suggestion, Figures had been checked and corrected with corresponding discussions in the revised manuscript. The experimental data quantifying cell proliferation and ECM deposition had been described and added with corresponding discussions.

- Line 477: “The dynamic reactor would improve cell proliferation dramatically”. Data is not provided.

As reviewer 3’s suggestion, the statement of “The dynamic reactor would improve cell proliferation dramatically” had been deleted to avoid confusion.

  1. Statistical analysis in missing in the manuscript.

 As reviewer 3’s suggestion, “Statistical analysis” had been added in the revised manuscript.

  1. The main conclusion(s) of the manuscript should be included in the abstract.

  As reviewer 3’s suggestion, main conclusion had been added in the abstract. The gelatin-based hydrogel scaffold encouraged chondrocytes proliferation promoting expression of collagen type II, aggrecan, and sox9 while retaining structural stability and durability of cartilage after dynamic compression and promote cartilage repair.

  1. Experimental methods: more details on the dynamic culture and parameters hydrogel stimulation should be added. 

  As reviewer 3’s suggestion, dynamic culture and parameters hydrogel stimulation had been added.

  1. Figures: confirm that all figures appear in proper locations in the text.

  As reviewer 3’s suggestion, all figures had been confirmed.

  1. The manuscript has several English spelling issues and requires revision by a native speaker. Some of the issues are detailed bellow.

 As reviewer 3’s suggestion,

- Line 16: “bubbles were corrected.” What this it means? 

 As reviewer 3’s suggestion, “bubbles were corrected.” had been corrected to be ” N2/gelatin bubbles were formed. “

- Line 21: “decomposition rate.” Authors wanted to say degradation?

 As reviewer 3’s suggestion, “decomposition rate.” had been corrected to be ” degradation”

- Line 24: “comfirm the active.” 

 As reviewer 3’s suggestion, “comfirm the active.”  had been corrected to be ” confirm the active.” .

- Line 26: “ccurture.”

 As reviewer 3’s suggestion, “ccurture.” had been corrected to be ” culture”.

- Line 59: “Gelatin was a mixture”

 As reviewer 3’s suggestion, “Gelatin was a mixture of proteins obtained by acid or alkaline hydrolysis of collagen.” had been modified to be” Gelatin was a protein, which was obtained by acid or base hydrolysis of collagen from an animal origin.”

- Line 83: “cartilage tissue engineering was important.” Important for what? 

 As reviewer 3’s suggestion, “cartilage tissue engineering was important.” had been deleted.

- Line 131: “hydrogel scaffold s”

 As reviewer 3’s suggestion,“hydrogel scaffold s” had been corrected to be ” hydrogel scaffolds”.

- Line 140: “cartilage was harvested from the rabbit knee joint was disinfected”

 As reviewer 3’s suggestion, “cartilage was harvested from the rabbit knee joint was disinfected” had been modified to be” The rabbit articular chondrocyte was isolated from the rabbit knee joint and disinfected initially with alcohol.”

- Line 457: “chondrocytes were planted in gelatin”

 As reviewer 3’s suggestion, “chondrocytes were planted in gelatin” had been modified to “chondrocytes were seeded on gelatin-based hydrogel scaffold (160μm)”

- Line 470: “staining 470 methods in section staining”

 As reviewer 3’s suggestion, “staining methods in section staining” had been modified to” Histological analysis such as hematoxylin and eosin staining are common staining methods.” In the revised manuscript.

- Line 473: “From the results in Figure 14, after seven days of static cell culture, which represents 473 cells’ localization in scaffolds”

 As reviewer 3’s suggestion, “From the results in Figure 14, after seven days of static cell culture, which represents 473 cells’ localization in scaffolds” had been modified to” From Figure 13, chondrocytes located at the scaffolds were observed after seven days of static cell culture.”

- Line 497: “Using the of microfluidic technology”

 As reviewer 3’s suggestion, “Using the of microfluidic technology” had been corrected to “Using the microfluidic technology”.

Reviewer 4 Report

The manuscript by Hsia-Wei Liu et al. titled: "Highly Organized Porous Gelatin-based Scaffold by Microfluidic 3D-foaming Technology and Dynamic Culture for Cartilage Tissue Engineering" describes a method of producing gelatin scaffolds with highly controlled porosity and examines biocompatibility of these scaffolds. The main merits of the manuscript are great precision on controlling monodisperse porosity and exploring the dynamic culture conditions that enable cell penetration into the scaffold. However, the manuscript has several major issues, that should be corrected:

1. There is no ethics statement on animal handling, even though researchers are harvesting and isolating chondrocytes from rabbit knee and using these primary cells in the biocompatibility testing.

2. The Materials & Methods chapter is missing some of the methods mentioned and presented in the Results section. This includes at least:

- Fixation of scaffolds after cell culture. Chapters 2.5-2.7 start with rehydration of histological slices, but the fixation of these slices is not described. For interpretation of results of this manuscript, also the direction of sectioning of histology slices would be important to mention.

- MTT assay and Live/Dead staining methods are not described. MTT results are also not shown, so was this done or not?

- DNA content, total collagen content and total GAG content measurement methods are not described.

- Selection of parameters for the dynamic cell culture in tidal bioreactor.

- Specifics of which antibodies were used for immunofluorescence staining. Also immunostainings were done on chondrocytes without scaffold as quality control to define their chondrocytic identity, these results are shown but methods not described.

- Details of uniform porous scaffold fabrication: Typical gas flow rate was 20-30μl/min and pressure 15-25 psi. So does this mean the lower values produce 100μm pores and higher values produce 160μm pores? This needs clarification.

- The source and type of gelatin used is not stated.

3. The Introduction could be improved by more specifically comparing gelatin to other biomaterials studied for osteoarthritis treatment. The references include rather exotic materials studied for this indication, how about simple Collagen in many forms? Likewise, the alternative chemical crosslinking strategies for gelatin should be introduced more, for example genipin, hydrazone crosslinking, photo/UV crosslinking etc.

4. The Discussion should compare presented results to existing literature, especially on the aforementioned different gelatin scaffolds and hydrogels.

5. The manuscript needs a language check/proof-reading.

Further minor issues and comments:

6. The authors used "vacuuming" as a term in production of the pores, would "degassing" be a more appropriate term?

7. Methods 2.2 lines 121-122: scaffold was soaked in water, removed the water, and weighted, then freeze-dried. Does this mean soaking sample was removed from the water, or was it dried first with another method before freeze-drying?

8. What is the "elasticity percentage" that is measure in mechanical testing and reported in Figure 4? How was this defined? Could you also show stress-strain curve from the mechanical testing (either a representative curve or an average curve)?

9. Line 149: which antibiotic was used here?

10. Chapter 3.2 discusses degradation behavior of hydrogel scaffolds, but fails to mention enzymatic degradation caused by the cells, which can be a significant factor for ECM-based hydrogels. This chapter could also mention the importance of porosity for cell migration and nutrient diffusion.

11. Is there literature reference for the used chondrocyte isolation method? Chondrocytes are known for slow proliferation, but they proliferate rather fast here, so it is important to verify their chondrocytic phenotype and make sure the cells are not fibroblasts.

12. The cell amount analysis from static cultures in Chapter 3.4. is confusing, some results indicate good proliferation and increase of cell amount, while others indicate cell death and reduction of cell amount.

13. Was there any statistical analysis done on the quantified data?

14. Figure 12 shows cross-section of scaffold after dynamic culture, but cross-section of static culture should be shown as well for comparison.

15. Line 444-446: "After dynamic culture, the number of cells increased, and type II collagen increased, but did not decrease, indicating that gelatin-based hydrogel scaffold (160μm) and chondrocytes did not de-differentiate after dynamic culture." What does this sentence mean?

16. The reference list could be updated with a few newer references, as currently most of the references are more than 5 years old.

Author Response

  1. There is no ethics statement on animal handling, even though researchers are harvesting and isolating chondrocytes from rabbit knee and using these primary cells in the biocompatibility testing.

As reviewer 4’s suggestion, statement of “Animal Ethics” had been added in the revised manuscript.  The information was supplied relating to ethical approvals: Animal study was approved by Institutional Animal Care and Use Committee of Master Laboratory Co. Ltd, Taiwan(IACUC Approval No.:20201210).

  1. The Materials & Methods chapter is missing some of the methods mentioned and presented in the Results section. This includes at least:

- Fixation of scaffolds after cell culture. Chapters 2.5-2.7 start with rehydration of histological slices, but the fixation of these slices is not described. For interpretation of results of this manuscript, also the direction of sectioning of histology slices would be important to mention.

As reviewer 4’s suggestion, the fixation of these slices had been described. The tissue/scaffold samples were processed according to standard histology procedures, being fixed in formalin (10%), dehydrated through alcohol and xylene passages, and embedded in a block of paraffin wax for slicing by a microtome(cross section).

- MTT assay and Live/Dead staining methods are not described. MTT results are also not shown, so was this done or not?

As reviewer 4’s suggestion, MTT assay and Live/Dead staining methods had been described in the revised manuscript. The cell viability and cell number were observed by live/dead assay and MTT assay. The cell viability was assessed using a live/dead viability/cytotoxicity assay kit and ethidium homodimer-1(EthD-1) and calcein AM were employed. After coculture, the hydrogel was rinsed with PBS buffer and stained with 500 μL live/dead assay each well. Images were observed using a fluorescence microscope (Olympus IX71, inverted microscope, Nagano, Japan). The live and dead cells were stained green or yellow, respectively.  For MTT assay, at different time points, one sample was taken from each culture and treated with 240 ?L of 3-(4,5-dimethylthiazolyl-2)-2,5-diphenyltetrazolium bromide (MTT, 0.5 mg/mL; Sigma) in Dulbecco’s modified eagle medium(DMEM)(Biowest) at 37oC for 4 h in dark. Insoluble formazan crystals reduced from MTT were extracted with dimethyl sulfoxide(DMSO). The absorbance (optical density (OD)) at 570 nm of the extractant was detected.

- DNA content, total collagen content and total GAG content measurement methods are not described.

 As reviewer 4’s suggestion, DNA content, total collagen content and total GAG content measurement methods had been described in the revised manuscript. The cell proliferation status was evaluated by DNA concentration analysis. Further-more, GAG and total collagen secretion were used to evaluate the degree of cell cartilage. The Quant-iTTM PicoGreen R dsDNA Reagent kit was used to detect the content of intra-cellular DNA. Quantitative analysis of deoxyribonucleic acid (DNA) was carried out by using fluorescence emission at 480 nm and 520 nm. The Blyscan Sulfated Glycosamino-glycan assay reagent combination was used to detect the content of soluble polyglucosa-mine in chondrocyte-hydrogel tissue. Quantitative analysis of polyglucosamine (GAG) was carried out by using Sepctrophotometer with a absorbance of 656  nm.  The Sircol Soluble Collagen Assay Kit was used to measure the total collagen content of each sample. The samples were measured by using immunofluorescence analyzer with a wavelength of 540 nm and calculated with a standard calibration line.

- Selection of parameters for the dynamic cell culture in tidal bioreactor.

  As reviewer 4’s suggestion, parameters for the dynamic cell culture in tidal bioreactor had been added.

- Specifics of which antibodies were used for immunofluorescence staining. Also immunostainings were done on chondrocytes without scaffold as quality control to define their chondrocytic identity, these results are shown but methods not described.  

 As reviewer 4’s suggestion, the statement of “immunofluorescence staining for chondrocytes without scaffold” had been added in the revised manuscript.  

- Details of uniform porous scaffold fabrication: Typical gas flow rate was 20-30 μl/min and pressure 15-25 psi. So does this mean the lower values produce 100μm pores and higher values produce 160μm pores? This needs clarification.

  As reviewer 4’s suggestion, the statement of “Microbubbles were generated in a focusing flow at input liquid flow=20 μl/min and air pressure=15 and 25 psi for microbubbles sizes of 100μm and 160μm, respectively[11,12]” had been added in the revised manuscript(Section 2.1).

- The source and type of gelatin used is not stated.

  As reviewer’s suggestion, “ Type A gelatin (Sigma-Aldrich, St. Louis, MO, USA)” had been added in the revised manuscript(Section 2.1).

  1. The Introduction could be improved by more specifically comparing gelatin to other biomaterials studied for osteoarthritis treatment. The references include rather exotic materials studied for this indication, how about simple Collagen in many forms? Likewise, the alternative chemical crosslinking strategies for gelatin should be introduced more, for example genipin, hydrazone crosslinking, photo/UV crosslinking etc.

  As reviewer 4’s suggestion, the alternative chemical crosslinking strategies for gelatin had been added in the revised manuscript.

  1. The Discussion should compare presented results to existing literature, especially on the aforementioned different gelatin scaffolds and hydrogels.

  1. The manuscript needs a language check/proof-reading.

 As reviewer 4’s suggestion, language had been checked.

Further minor issues and comments:

  1. The authors used "vacuuming" as a term in production of the pores, would "degassing" be a more appropriate term?

 As reviewer 4’s suggestion, “vacuuming" had been corrected to "degassing".

  1. Methods 2.2 lines 121-122: scaffold was soaked in water, removed the water, and weighted, then freeze-dried. Does this mean soaking sample was removed from the water, or was it dried first with another method before freeze-drying?

As reviewer 4’s suggestion, “ dried first with another method before freeze-drying” had been added and the statement of “scaffold was soaked in water, removed the water, and weighted, then freeze-dried.” had been modified to the statement of “ The gelatin-based hydrogel scaffold was soaked in water, confirmed that all pores of the scaffold was filled with water, wiped with wet tissue paper to remove surface droplets, and weighted as Ws.”. 

  1. What is the "elasticity percentage" that is measure in mechanical testing and reported in Figure 4? How was this defined? Could you also show stress-strain curve from the mechanical testing (either a representative curve or an average curve)?

As reviewer 4’s suggestion, the results of "elasticity percentage" reported in Figure 4 was deleted to avoid confusion in the revised manuscript.

  1. Line 149: which antibiotic was used here?

As reviewer 4’s suggestion, the antibiotics containing Penicillin and Streptomycin had been added in the revised manuscript.

  1. Chapter 3.2 discusses degradation behavior of hydrogel scaffolds, but fails to mention enzymatic degradation caused by the cells, which can be a significant factor for ECM-based hydrogels. This chapter could also mention the importance of porosity for cell migration and nutrient diffusion.

  1. Is there literature reference for the used chondrocyte isolation method? Chondrocytes are known for slow proliferation, but they proliferate rather fast here, so it is important to verify their chondrocytic phenotype and make sure the cells are not fibroblasts.

 As reviewer 4’s suggestion, reference 27 for the used chondrocyte isolation method was added in the revised manuscript. The cytoskeleton-specific proteins such as collagen type II, Sox 9, Aggrecan, MMP13 were stained by indirect immunofluorescence staining as shown in Figure 5. The statement “Cells isolated from rabbit knee cartilage were labeled with collagen type II, aggrecan, MMP13, and Sox9 by fluorescent staining which were cartilage-specific proteins for identification of chondrocytes as shown in Figures 5(C)-5(F).” was added in the revised manuscript.

  1. The cell amount analysis from static cultures in Chapter 3.4. is confusing, some results indicate good proliferation and increase of cell amount, while others indicate cell death and reduction of cell amount.

 As reviewer 4’s suggestion, description of the results from static cultures had been modified to avoid confusion. Cell viability and cell growth could be observed by live/dead fluorescence staining. From Figure 6, living cells were stained with green fluorescence and dead cells were stained with yellow fluorescence. The fluorescent images showed live (green) chondrocytes cultured on gelatin-based hydrogel scaffolds over a static culturing period of 7 days. The total number of cells increased with the days. However, most of cells were dead cells and the numbers of dead cells were increased with days. That is, the numbers of live cells decreased with days which might be due to the cellularity and localization of chondrocytes on gelatin-based hydrogel scaffolds. After static culturing, the chondrocytes would be covered on the most part of gelatin-based hydrogel scaffolds’ surface over a period of 21 days. The chondrocytes-covered gelatin-based hydrogel scaffolds exhibited low interconnectivity in static culture system which might be resulted in the observation of dead cells. Further, the cell viability and cell number could be detected by MTT assay.  From results of MTT assay, the cell numbers gradually decreased with days which was similar to the re-sults of live/dead assay.  The living cell numbers decreased from 2.3x105 cells(after 1 day) to 4.6x104 cells(after 21 days).

  1. Was there any statistical analysis done on the quantified data?

 As reviewer 4’s suggestion, the statement of statistical analysis had been added.

  1. Figure 12 shows cross-section of scaffold after dynamic culture, but cross-section of static culture should be shown as well for comparison.

As reviewer 4’s suggestion, the cross-section of static culture had been added in the Figures 9(C)and (D) of revised manuscript.

  1. Line 444-446: "After dynamic culture, the number of cells increased, and type II collagen increased, but did not decrease,indicating that gelatin-based hydrogel scaffold (160μm) and chondrocytes did not de-differentiate after dynamic culture." What does this sentence mean?

As reviewer 4’s suggestion, the statement “After dynamic culture, the number of cells increased, and type II collagen increased, but did not decrease, indicating that gelatin-based hydrogel scaffold (160μm) and chondrocytes did not de-differentiate after dynamic culture.”  had been modified to “After dynamic culture, the number of cells and type II collagen increased, indicating that differentiation of chondrocytes was carried out on gelatin-based hydrogel scaffold (160μm).”

  1. The reference list could be updated with a few newer references, as currently most of the references are more than 5 years old.

As reviewer 4’s suggestion, some new references had been added.

20.Huang, C.C.; Chen, Y.J.; Liu, H.W. Characterization of Composite Nano-Bioscaffolds Based on Collagen and Supercritical Fluids-Assisted Decellularized Fibrous Extracellular Matrix, Polymers, 2021; Volume 13, pp.4326.

25.Kushibiki, T; Mayumi, Y;Nakayama, E; Azuma, R; Ojima, K; Horiguchi, A; Ishihara, M. Photocrosslinked gelatin hydrogel improves wound healing and skin flap survival by the sustained release of basic fibroblast growth factor. Sci Rep., 2021; Volume11(1), p. 23094.

26.Gupta, D.; Santoso, J.W.; McCain, M.L. Characterization of Gelatin Hydrogels Cross-Linked with Microbial Transglutaminase as Engineered Skeletal Muscle Substrates. Bioengineering, 2021; Volume 8, p.6.

Round 2

Reviewer 3 Report

Critical issues regarding the characterization of hydrogels and the quantification of ECM were not properly addressed. Please see my comments bellow.

1. Authors have deleted data regarding the mechanical properties (Fig. 4 in the first version). As previously suggested, such data is relevant for the manuscript and should be repeated and included in the text. 

2. It remains unclear whether authors have addressed the following points in the revised version.

- A major issue concerns to the quantification of ECM. I have raised my concerns to that in point 7 (previous review report). However, such issues have not been properly addressed in the revised version. For instance, authors state that “After dynamic culture, the number of cells and type II collagen increased”. How do the authors quantify the cell number and collagen content? There is no quantitative data supporting such claim. Please address the point 7 in my previous review report.

- Statistical analysis remains absent in the graphs. Did the authors perform statistical analysis or the data has no statistically significant differences?

Author Response

  1. Authors have deleted data regarding the mechanical properties (Fig. 4 in the first version). As previously suggested, such data is relevant for the manuscript and should be repeated and included in the text. 

    As reviewer 3’s suggestion, data regarding the mechanical properties had been added and the modified descriptions had also been added in the revised manuscript as following:

“The tensile stress values of EDC-crosslinked gelatin hydrogels with different pore sizes were determined to be ca.10.8 and 8.4 for 100μm and 160μm, respectively as shown in Figure 4(C) which indicated structural strength of the resulting EDC-crosslinked gelatin-based hydrogel scaffolds with different pore sizes(160 and 100 μm).  Further, the elongations at break of EDC-crosslinked gelatin-based hydrogel scaffolds with different pore sizes(160 and 100 μm) were observed in a range of 19~22%., indicating the EDC-crosslinked gelatin-based hydrogel scaffolds with different pore sizes(160 and 100 μm) all exhibited flexibility[Figure 4(C)].” 

  1. It remains unclear whether authors have addressed the following points in the revised version.

- A major issue concerns to the quantification of ECM. I have raised my concerns to that in point 7 (previous review report). However, such issues have not been properly addressed in the revised version. For instance, authors state that “After dynamic culture, the number of cells and type II collagen increased”. How do the authors quantify the cell number and collagen content? There is no quantitative data supporting such claim. Please address the point 7 in my previous review report.

     As reviewer 3’s suggestion, the data for the statement of “After dynamic culture, the number of cells and type II collagen increased” had been added such as DNA content, collagen/DNA, GAG/DNA as shown in Figure 12 of manuscript. The results and discussions had also been described in the revised manuscript. 

- Statistical analysis remains absent in the graphs. Did the authors perform statistical analysis or the data has no statistically significant differences?

 As reviewer 3’s suggestion, results of statistical analysis were added in the revised manuscript. Values with different superscripts were significantly different (P < 0.05);* , statistical significance between the groups (p<0.05); +, nonsignificance within the group (p>0.05). The “*” and “+” had bee added in the revised Figures.

Reviewer 4 Report

The revised manuscript by Hsia-Wei Liu et al. titled: "Highly Organized Porous Gelatin-based Scaffold by Microfluidic 3D-foaming Technology and Dynamic Culture for Cartilage Tissue Engineering" has been significantly improved by the round of major revision. The reviewer questions have mostly been answered satisfactorily, but there are still issues related to this manuscript that require further attention.

1. The authors claim that proof-reading and language check has been done on the manuscript, but it still contains lots of grammatical errors or typos. Especially the Introduction also has weird change between past tense forms, i.e. what was done or what is done, making it seems that some of the stated issues have been already solved previously.

2. The Discussion part of the manuscript comparing to other published works is still inadequate. This was now added during revision to chapter 3.3., but not from 3.4. on-wards.

3. The Live/Dead results and comparison to MTT assay are still confusing. In Figure 6. A, B, C you can see increase of all cells, even though the text mentions decrease of live cells and increase of only dead cells. At the same time, the MTT assay shows clear decrease in cell number, but to my understanding this could actually be just decrease in cell activity, not exactly decrease in cell number. Furthermore, Live/Dead stainings are often shown as green/red, instead of yellow, to get higher contrast between the live and dead cells. Can the colors be changed in the microscope images for this more conventional coloring?

4. The mechanical properties were measured, but in revision the results were removed. I recommend including these results, but not as elasticity%, but just as the average stress-strain curves.

5. The Materials and Methods section still does not include all the producers of used reagents. Likewise, the parameters (rpm?) used with tidal bioreactor are also still missing.

6. The use of abbreviations should be re-checked. For example, the use of GAG before it is defined as glycosaminoglycan. There is also a mention of "glucaminoglycan" and "polyglucosamine", are these separate from glucosaminoglycan (GAG)?. Also, after the abbreviation is defined, then please use this abbreviation instead of the whole word (for example chapter 3.6.).

7. This excerpt from previous revision round: "14. Figure 12 shows cross-section of scaffold after dynamic culture, but cross-section of static culture should be shown as well for comparison.

As reviewer 4’s suggestion, the cross-section of static culture had been added in the Figures 9(C)and (D) of revised manuscript."

The authors response gives the impression that something was added to the Figure 9, but in reality only the figure caption was modified. This is misleading. The point of that comment was to add comparison between static and dynamic culture cross-sections in (current) Figure 11.

8. Chapter 2.8. about statistical analysis was added to the manuscript, but none of the figures have an asterisk marking statistical significance (Fig. 4, Fig. 6, Fig. 7, Fig. 8)

9. Chapter 2.6 on lines 252-258 seems to be misplaced or missnumbered.

Minor issues:

- Abstract and possibly other parts say "... introducing an appropriate nitrogen gas and gelatin solution at an appropriate flow rate...", could this be "optimized" instead of "appropriate", especially if these parameters were optimized by the authors.

- Not all instances of replacing "vacuuming" with "degassing" have been done.

- Several instances mention optical microscope and confocal microscope, but confocal is also an optical microscope, opposed to electron microscope. Perhaps the authors mean bright field microscope and confocal microscope?

- Line 129: "conjugated cross microscopy", should this be "conjugated focus microscopy" instead? If not, then please elaborate more on the technique.

- Line 421-423: "That is, the numbers of live cells decreased with days which might be due to the cellularity and localization of chondrocytes on gelatin-based hydrogel scaffolds." This sentence is confusing.

- Line 544-545: "...dynamic culture time indicated that the degree of chondrocyte differentiation of chondrocytes increases." This sentence is confusing.

Author Response

Reviewer 4

  1. The authors claim that proof-reading and language check has been done on the manuscript, but it still contains lots of grammatical errors or typos. Especially the Introduction also has weird change between past tense forms, i.e. what was done or what is done, making it seems that some of the stated issues have been already solved previously.

 As reviewer4’s suggestions, the proof-reading and language check must be done on the manuscript. I had tried my best and looked for native speaker’ help to check it.

  1. The Discussion part of the manuscript comparing to other published works is still inadequate. This was now added during revision to chapter 3.3., but not from 3.4. on-wards.

  As reviewer4’s suggestions, the Discussion part of the manuscript comparing to other published works had been added in the 3.3 of revised manuscript.

  1. The Live/Dead results and comparison to MTT assay are still confusing. In Figure 6. A, B, C you can see increase of all cells, even though the text mentions decrease of live cells and increase of only dead c ells. At the same time, the MTT assay shows clear decrease in cell number, but to my understanding this could actually be just decrease in cell activity, not exactly decrease in cell number. Furthermore, Live/Dead stainings are often shown as green/red, instead of yellow, to get higher contrast between the live and dead cells. Can the colors be changed in the microscope images for this more conventional coloring?

 As reviewer 4’s suggestion, the description of Live/Dead results and MTT assay had been modified to avoid confusing.  As reviewer 4’s suggestion, he MTT assay showed clear decrease in cell number, indicating a decrease trend in cell activity. The active cell numbers decreased from 2.3x105 cells(after 1 day) to 4.6x104 cells(after 21 days). The total number of cells containing dead and live cells increased with the days. However, the MTT assay showed clear decrease in cell number.  After static culturing, the chondrocytes were covered on the most part of EDC-crosslinked gelatin-based hydrogel scaffolds’ surface over a period of 21 days as shown in Figure 6(C).

 As reviewer 4’s suggestion, the description had been corrected and reference had been cited. Living cells were stained with green fluorescence and dead cells were stained with red fluorescence. The fluorescent images appeared yellow in overlay overlap of live and dead cells.

  1. The mechanical properties were measured, but in revision the results were removed. I recommend including these results, but not as elasticity%, but just as the average stress-strain curves.

  As reviewer 4’s suggestion, the mechanical properties such as tensile stress and elongation had been added in the revised manuscript.

  1. The Materials and Methods section still does not include all the producers of used reagents. Likewise, the parameters (rpm?) used with tidal bioreactor are also still missing.

 As reviewer 4’s suggestion, all the producers of used reagents had been described in the section2.1 of revised manuscript. All the chemicals, such as Type A gelatin(Sigma-Aldrich, USA), Pluronic F127(Invitrogen, UK), perfluorohexane(Merck,Germany), EDC[1-ethyl-3-(3-dimethylaminopropyl) carbodiimide hydrochloride](Merck,Germany), phosphate buffered saline(PBS)(Merck, Germany), Quant-iT™ PicoGreen® dsDNA Assay Kit (Invitrogen,UK), Sircol soluble collagen Assay Kit (Biocolor, UK), Blyscan Sulfated Glycosaminoglycan assay Kit,  proteinase K  (Merck,Germany), Dulbecco’s Phosphate Buffered Saline (DPBS)(Sigma-Aldrich, USA), Dulbecco’s modified eagle’s medium (DMEM) (Biowest, France), dimethyl sulfoxide(DMSO)(Merck,Germany), 3-(4,5-cimethylthiazol-2-yl)-2,5-diphenyltetrazolium bromide, MTT(Merck ,Germany), live/dead viability/cytotoxicity assay kit (Thermo Fisher Scientific, USA), ethidium homodimer-1(EthD-1)(Biotium, USA), Tween 20 (Sigma-Aldrich, USA), MES(J.T Baker, USA), Eosin(Merck,Germany), alcian blue(Merck,Germany), hematoxylin(Merck ,Germany), Triton X-100(Merck,Germany), xylene(Merck,Germany), eosin-phloxine. antibiotics(Penicillin-Streptomycin;10,000 U/mL)(Thermo Fisher Scientific, USA), collagenase (Sigma-Aldrich, USA), calcein AM(Thermo Fisher Scientific, USA), were used without further purification.

 As reviewer 4’s suggestion, the parameters used with tidal bioreactor had been added. The incubator temperature was 37 ℃, the volumetric concentration 5% of CO2. The rotating speed was 150 rpm.

  1. The use of abbreviations should be re-checked. For example, the use of GAG before it is defined as glycosaminoglycan. There is also a mention of "glucaminoglycan" and "polyglucosamine", are these separate from glucosaminoglycan (GAG)?. Also, after the abbreviation is defined, then please use this abbreviation instead of the whole word (for example chapter 3.6.).

 As reviewer 4’s suggestion, abbreviation(GAG) had been instead of the whole word (for example chapter 3.6.).

  1. This excerpt from previous revision round: "14. Figure 12 shows cross-section of scaffold after dynamic culture, but cross-section of static culture should be shown as well for comparison.

As reviewer 4’s suggestion, the cross-section of static culture had been added in the Figures 9(C)and (D) of revised manuscript."

The authors response gives the impression that something was added to the Figure 9, but in reality only the figure caption was modified. This is misleading. The point of that comment was to add comparison between static and dynamic culture cross-sections in (current) Figure 11.

As reviewer 4’s suggestion, comparison between static and dynamic culture cross-sections had been added in revised Figure 11. However, SEM morphology of chondrocyte grown on EDC-crosslinked gelatin-based hydrogel scaffold after static culture for 28 days(cross section) is difficult to provide. The proposed drawings were modified for comparison between static and dynamic culture. The statement for comparison between static and dynamic culture was added in the revised manuscript as following:

“Because poor migration of cells was observed after static culture for 14 days, static culture was displaced with dynamic culture to promote migration of cells. Poor cell migration after static culture might be exhibited and good migration of cells after dynamic culture could be observed for 28 days.  Viewed from the cross section of scaffold, chondrocytes had been migrated, uniformly distributed, and proliferated in the EDC-crosslinked gelatin-based hydrogel scaffold after 28 days of dynamic culture [Figures 11(A)]. Comparison between static and dynamic culture were proposed as shown in Figures 11(B) and 11(C), respectively. “

  1. Chapter 2.8. about statistical analysis was added to the manuscript, but none of the figures have an asterisk marking statistical significance (Fig. 4, Fig. 6, Fig. 7, Fig. 8)

  As reviewer 4’s suggestion, the asterisk marking statistical significance had been added.

  1. Chapter 2.6 on lines 252-258 seems to be misplaced or missnumbered.

 As reviewer 4’s suggestion, Chapter 2.6 had been corrected to be 2.9.

Minor issues:

- Abstract and possibly other parts say "... introducing an appropriate nitrogen gas and gelatin solution at an appropriate flow rate...", could this be "optimized" instead of "appropriate", especially if these parameters were optimized by the authors.

  As reviewer 4’s suggestion, "... introducing an appropriate nitrogen gas and gelatin solution at an appropriate flow rate..." had been corrected to be " introducing an optimized nitrogen gas and gelatin solution at an optimized flow rate ".

- Not all instances of replacing "vacuuming" with "degassing" have been done.

  As reviewer 4’s suggestion, "vacuuming" had been replaced with "degassing" in the revised manucript.

- Several instances mention optical microscope and confocal microscope, but confocal is also an optical microscope, opposed to electron microscope. Perhaps the authors mean bright field microscope and confocal microscope?

As reviewer 4’s suggestion, “optical microscope and confocal microscope" had been corrected to” bright field microscope and confocal microscope”

- Line 129: "conjugated cross microscopy", should this be "conjugated focus microscopy" instead? If not, then please elaborate more on the technique.

As reviewer 4’s suggestion, "conjugated cross microscopy" had been corrected to "conjugated focus microscopy".

- Line 421-423: "That is, the numbers of live cells decreased with days which might be due to the cellularity and localization of chondrocytes on gelatin-based hydrogel scaffolds." This sentence is confusing.

As reviewer 4’s suggestion,"That is, the numbers of live cells decreased with days which might be due to the cellularity and localization of chondrocytes on gelatin-based hydrogel scaffolds."  had been deleted to avoid confusing in the revised manuscript.

- Line 544-545: "...dynamic culture time indicated that the degree of chondrocyte differentiation of chondrocytes increases." This sentence is confusing.

As reviewer 4’s suggestion,"...dynamic culture time indicated that the degree of chondrocyte differentiation of chondrocytes increases." had been deleted to avoid confusing in the revised manuscript.

Round 3

Reviewer 4 Report

The revised manuscript by Hsia-Wei Liu et al. titled: "Highly Organized Porous Gelatin-based Scaffold by Microfluidic 3D-foaming Technology and Dynamic Culture for Cartilage Tissue Engineering" has been improved by the round of revision again. However, some issues persist:

1. The manuscript still needs a language check/proof-reading. If no native English-speakers are available at the authors institute, I recommend commercial language check services. Even MDPI has language check service for authors.

2. The mechanical testing results still need a bit of clarification. From what point of compression test is the "tensile stress" now recorded? Is it actually stress at break i.e. tensile strength? And should it be actually called "compressive stress/strength" and not "tensile stress/strength"?

3. The listing of used reagents in 2.1. is good, but it is still missing some reagents, at least immunofluorescence staining reagents Hoechst 33258, Dako Fluorescent Mounting Medium and both primary and secondary antibodies for collagen type II, Sox9, aggrecan, MMP13, and α-Actin.

4. The text has some redundancies (that would likely be also fixed in language check), for example: lines 130 & 135-137, line 411, lines 435 &436.

5. Figure 5 (A) caption says 100x, this is likely a typo.

6. Line 584 refers to Figure 12, but probably should refer to Figure 13.

Author Response

Dear Sir,

Thank you very much for your valuable comments and suggestions on my manuscript.

Enclosed, please find our copy of the revised manuscript entitled “Highly Organized Porous Gelatin-based Scaffold by Microfluidic 3D-foaming Technology and Dynamic Culture for Cartilage Tissue Engineering(ijms-1786577)”.  Amendments and changes were made as suggested by Reviewer.  We hope the revised version will now be suitable for publication in “Int. J. Mol. Sci.”.  The point-to-point response to the reviewers is enclosed in this letter.

  1. The manuscript still needs a language check/proof-reading. If no native English-speakers are available at the authors institute, I recommend commercial language check services. Even MDPI has language check service for authors.

 As reviewer4’s suggestion, MDPI’s language check service for authors had been considered.

  1. The mechanical testing results still need a bit of clarification. From what point of compression test is the "tensile stress" now recorded? Is it actually stress at break i.e. tensile strength? And should it be actually called "compressive stress/strength" and not "tensile stress/strength"?

As reviewer4’s suggestion, the description of ”Mechanical property of EDC-crosslinked gelatin-based hydrogel scaffold” had been corrected.  The original statement of “Mechanical property of EDC-crosslinked gelatin-based hydrogel scaffold was determined by TA.XT-plus Texture analyzer such as elasticity……..be used to calculate the elasticity of the gelatin-based hydrogel scaffold.” for elasticity had been corrected to be those for tensile strength and elongation at break. 

The tensile test was then performed by pulling off the EDC-crosslinked gelatin-based hydrogel scaffold at pretest speed test, test speed, post test speed of 1, 1 and 10 mm/s using a texture analyzer (TA.XT-plus Texture analyzer, UK). The net length between the jaws was almost constant for all films and 20 mm. The texture analyzer was run at auto force mode with the trigger force of 5 g. From stress–strain curves two parameters were calculated: tensile strength was calculated as maximum stress and elongation at break where the film was torn.

  1. The listing of used reagents in 2.1. is good, but it is still missing some reagents, at least immunofluorescence staining reagents Hoechst 33258, Dako Fluorescent Mounting Medium and both primary and secondary antibodies for collagen type II, Sox9, aggrecan, MMP13, and α-Actin.

As reviewer4’s suggestion, immunofluorescence staining reagents Hoechst 33258, Dako Fluorescent Mounting Medium and both primary and secondary antibodies for collagen type II, Sox9, aggrecan, MMP13, and α-Actin had been added in the revised manuscript(2.1).

The immunofluorescence staining reagents containing Hoechst 33258(Sigma-Aldrich, USA), Dako Fluorescent Mounting Medium(Sigma-Aldrich, USA)), anti-collagen II primary antibody(Sigma-Aldrich, Germany), anti-sox9 primary antibody(Sigma-Aldrich, Germany), anti-aggrecan primary antibody (Thermo Fisher Scientific, USA), anti-MMP13 primary antibody (Sigma-Aldrich, Germany), anti-α-smooth muscle actin-Cy3 antibody (anti-α-SMA-Cy3)(Sigma-Aldrich, Germany), and Goat anti-Mouse IgG (H+L) secondary antibody, Alexa Fluor 488(Thermo Fisher Scientific, USA) were employed.

  1. The text has some redundancies (that would likely be also fixed in language check), for example: lines 130 & 135-137, line 411, lines 435 &436.

 As reviewer4’s suggestion, the statement “Typically, the liquid flow rate was 20–30 μl/min and the air pressure was 15–25 psi[11,12].” had been deleted in the revised manuscript(lines 130 & 135-137).  The statement “ After static culturing, the chondrocytes were covered on the most part of EDC-crosslinked gelatin-based hydrogel scaffolds’ surface over a period of 21 days as shown in Figure 6(C). The chondrocytes-covered EDC-crosslinked gelatin-based hydrogel scaffolds exhibited low interconnectivity in static culture system which might be overcrowded on surface of scaffold. “ had been modified to“After static culturing, the chondrocytes were covered on the most part of EDC-crosslinked gelatin-based hydrogel scaffolds’surface over a period of 21 days as shown in Figure 6(C) which exhibited overcrowded chondrocytes on surface of scaffold and low interconnectivity. “ .

  1. Figure 5 (A) caption says 100x, this is likely a typo.

 As reviewer4’s suggestion, the “100x” had been deleted in the revised manuscript.

  1. Line 584 refers to Figure 12, but probably should refer to Figure 13.

 As reviewer4’s suggestion, the “Figure 12” had been corrected to “Figure 13” in the revised manuscript.

Best regards,

Ching-Cheng Hunag
